# Macrophage dysfunction initiates colitis during weaning of infant mice lacking the interleukin-10 receptor

Naresh S Redhu[1,2], Vasudevan Bakthavatchalu[3], Evan A Conaway[4], Dror S Shouval[1,5,6], Amy Tsou[1,2], Jeremy A Goettel[1,2], Amlan Biswas[1,2], Chuanwu Wang[3], Michael Field[1], Werner Muller[7], Andre Bleich[8], Ning Li[9], Georg K Gerber[9], Lynn Bry[9], James G Fox[3], Scott B Snapper[1,2,10]*, Bruce H Horwitz[2,4]*

[1]Division of Gastroenterology, Hepatology and Nutrition, Boston Children's Hospital, Boston, United States; [2]Harvard Medical School, Boston, United States; [3]Division of Comparative Medicine, Massachusetts Institute of Technology, Cambridge, United States; [4]Department of Pathology, Brigham and Women's Hospital, Boston, United States; [5]Division of Pediatric Gastroenterology and Nutrition, Edmond and Lily Safra Children's Hospital, Sheba Medical Center, Tel Hashomer, Israel; [6]Sackler Faculty of Medicine, Tel Aviv University, Tel Aviv, Israel; [7]Faculty of Biology, Medicine and Health, University of Manchester, Manchester, United Kingdom; [8]Institute for Laboratory Animal Science and Central Animal Facility, Hannover Medical School, Hannover, Germany; [9]Department of Pathology, Massachusetts Host-Microbiome Center, Brigham and Women's Hospital, Boston, United States; [10]Division of Gastroenterology, Brigham and Women's Hospital, Boston, United States

*For correspondence: Scott. Snapper@childrens.harvard.edu (SBS); bhorwitz@partners.org (BHH)

Competing interests: The authors declare that no competing interests exist.

**Abstract** Infants with defects in the interleukin 10 receptor (IL10R) develop very early onset inflammatory bowel disease. Whether IL10R regulates lamina propria macrophage function during infant development in mice and whether macrophage-intrinsic IL10R signaling is required to prevent colitis in infancy is unknown. Here we show that although signs of colitis are absent in IL10R-deficient mice during the first two weeks of life, intestinal inflammation and macrophage dysfunction begin during the third week of life, concomitant with weaning and accompanying diversification of the intestinal microbiota. However, IL10R did not directly regulate the microbial ecology during infant development. Interestingly, macrophage depletion with clodronate inhibited the development of colitis, while the absence of IL10R specifically on macrophages sensitized infant mice to the development of colitis. These results indicate that IL10R-mediated regulation of macrophage function during the early postnatal period is indispensable for preventing the development of murine colitis.

## Introduction

Inflammatory bowel diseases (IBD) including Crohn's disease (CD) and ulcerative colitis (UC) are thought to result from the interaction of microbial and environmental factors with a genetically susceptible host. While IBD susceptibility is generally considered to arise from complex genetic risk in adolescent and adult patients, those who develop symptoms during infancy (age <2), or before 6 years of age (very early onset IBD, VEOIBD), often present with disease refractory to conventional

**eLife digest** Inflammation is an immune response that helps the body to repair damaged tissues and defend itself against bacteria and other harmful microbes. Immune cells called macrophages stimulate inflammation when they detect bacteria and other microbes. However, the strength of the inflammatory response is tightly controlled to prevent too much inflammation when it is not necessary. There is evidence that an inability to control the activity of macrophages can lead to excessive inflammation that can in itself cause injury.

Inflammatory bowel diseases, including Crohn's disease and ulcerative colitis, occur when the intestine becomes inflamed in response to the bacteria that normally live there. Most patients with these diseases first present symptoms as young adults, but in rare cases the symptoms appear during infancy. Some patients who develop symptoms early in life carry a mutation in a gene encoding IL10R, a receptor protein that normally inhibits the ability of macrophages to cause inflammation. However, it was not known exactly when the inflammation begins in infants.

Mice are often used in research as models of human health and disease. Redhu et al. investigated the role of IL10R in the intestines of infant mice. These experiments showed that the intestines of mutant mice that lacked the IL10R protein became inflamed as they were weaned from breast milk to a solid diet. This inflammation was accompanied by a greater increase in the number of actively recruited macrophages in the intestine of mutant mice compared to normal mice. Further experiments revealed that macrophages in the mutant mice activate many genes involved in inflammation. The transition from breast milk to a solid diet is accompanied by large increases in intestinal bacteria. Treating the mice with antibiotics decreased the number of bacteria in the intestines and reduced the level of inflammation, as did treating the mice with a compound that killed macrophages.

The findings of Redhu et al. suggest that IL10R prevents macrophages from causing inflammation when infant mice are weaned. This, in turn, suggests that treatments that modify intestinal macrophages, as well as bacteria, could help infants that have, or are at risk of developing inflammatory bowel disease. In the longer-term, these findings might also aid the development of new treatments for older patients with inflammatory bowel diseases.

therapies and are more likely to have disease-causative single gene mutations (*Uhlig et al., 2014*). A detailed understanding of disease pathogenesis in infants, as well as model systems that recapitulate infantile-onset IBD, may lead to fundamental insights into the etiology of IBD, and identify novel therapeutic strategies. In this regard, rare loss-of-function mutations in *IL10*, or its receptors *IL10RA* or *IL10RB* in humans lead to infantile onset IBD (*Glocker et al., 2010*, *2009*; *Moran et al., 2013*; *Shouval et al., 2014a*; *Kotlarz et al., 2012*), suggesting that the IL10/IL10R pathway is essential for intestinal mucosal homeostasis and prevention of intestinal inflammation in infancy. Thus, modeling infantile-onset colitis in mice that lack functional IL10R signaling could lead to significant advances in our understanding of the etiopathogenesis of these diseases.

IL10 is a potent anti-inflammatory cytokine that inhibits proinflammatory and co-stimulatory function in both innate and adaptive immune cells (*Ding and Shevach, 1992*; *Fiorentino et al., 1991*; *Saraiva and O'Garra, 2010*). Like humans, mice deficient in IL10 or the IL10R on the 129SvEv background develop chronic spontaneous colitis (*Kang et al., 2008*; *Kühn et al., 1993*; *Spencer et al., 1998*), strongly supporting the concept that IL10 signaling plays a central role in preventing mucosal inflammation. These murine models have enabled investigations into the cellular and microbial factors that contribute to the development of IBD. However, because colitis in these models presents after an extended latency, understanding the developmental events early in life that contribute to disease development remain to be defined.

The IL10 receptor (IL10R) is a heterotetramer consisting of two chains of ligand-binding IL10R$\alpha$ (IL10R1) and two subunits of signaling IL10R$\beta$ (IL10R2) (*Donnelly et al., 2004*; *Moore et al., 2001*). The IL10R is expressed on cells of both the innate and adaptive immune compartments and a number of studies have highlighted the role of IL10R on subsets of regulatory T cells and T helper 17 (Th17) cells in preventing the development of colitis (*Chaudhry et al., 2011*; *Huber et al., 2011*;

*Kamanaka et al., 2011*; *Murai et al., 2009*). Recently, we and others have demonstrated a requirement for IL10R in innate immune cells to prevent murine colitis (*Li et al., 2015*; *Shouval et al., 2014b*, *2016*; *Zigmond et al., 2014*). In addition, we have shown excessive accumulation of inflammatory Ly6C$^+$ monocytes within the colons of colitic IL10R-deficient mice as well as defects in the differentiation of both murine and human anti-inflammatory macrophages (MΦs) defective in IL10R signaling (*Shouval et al., 2014b*). It has previously been shown that intestinal microbiota promotes the accumulation of Ly6C$^+$ monocytes within the lamina propria (LP) (*Bain et al., 2014*) and there is ample evidence that the microbiota is intimately involved in the development of colitis in many susceptible mouse models including mice lacking IL10 signaling (*Kang et al., 2008*; *Sellon et al., 1998*). Taken together, these studies imply that IL10R signaling on intestinal MΦs prevents microbially-driven recruitment of Ly6C$^+$ monocytes and the development of intestinal inflammation.

While the studies outlined above demonstrate the critical role of IL10R signaling within MΦs in preventing inflammatory responses in the large intestine, the colitis that develops spontaneously in mice that are completely-deficient in IL10R or that specifically lack IL10R within CX3CR1$^+$ MΦs manifest with a prolonged latency of several weeks, and is strain- and microbiota-dependent (*Kang et al., 2008*; *Kühn et al., 1993*; *Spencer et al., 1998*; *Zigmond et al., 2014*). Given that colitis in infants lacking functional IL10R is an aggressive disease that presents very early in life (*Glocker et al., 2009*; *Shouval et al., 2014a*), it is unclear whether the same cellular mechanisms observed in adult mice lacking IL10R are in fact operating within the time frame of infant development. Furthermore, it has not been determined whether specific developmental events during the post-partum period such as initial bacterial colonization at the time of birth, or the diversification of the infant microbiome that occurs at the time of weaning are influenced by the absence of IL10R and/or play catalytic roles in driving inflammation in the absence of IL10R.

To address these issues, we performed detailed time-series experiments using IL10R-deficient infant mice and their littermate co-housed controls, and evaluated histology, inflammatory gene expression, cellular composition of the LP, and luminal bacterial communities from birth to 14 weeks of age. We used clodronate depletion to investigate the role of colonic MΦs in colonic inflammation that develops in infant IL10R-deficient mice. In addition, we analyzed the onset of colitis in crosses between C57BL/6 mice lacking IL10Rα specifically in MΦs and those carrying a genetic locus driving colitis susceptibility, *Cdcs1*, neither of which independently develop colitis. Our data demonstrate that the requirement for macrophage-intrinsic IL10R signaling develops during a critical developmental transition, which has important implications for understanding factors that initiate colitis in infants with genetic deficiencies in IL10R signaling.

## Results

### Colitis develops in 3–4 week old *Il10rb*$^{-/-}$ mice

To identify the onset of colitis in IL10R deficiency, we examined the kinetics of intestinal inflammation from early post-natal life through adulthood in *Il10rb*$^{-/-}$ mice (129SvEv background) under specific pathogen-free (SPF) conditions. As shown by H and E staining of distal colonic sections in *Figure 1A*, colonic inflammatory infiltrates could be detected in *Il10rb*$^{-/-}$ but not littermate *Il10rb*$^{+/-}$ (control) mice by the third week of life. By the fourth week, more than 80% of *Il10rb*$^{-/-}$ mice exhibited moderate colitis with crypt abscesses and crypt hyperplasia (*Figure 1A,B*), and virtually all mice displayed histologic signs of colitis by 8 weeks of age. At the end of this time-series (30 weeks), 100% of *Il10rb*$^{-/-}$ mice developed severe subacute diffuse colitis (*Figure 1A,B*). At this age, endoscopic imaging and clinical scores revealed severe colitis in *Il10rb*$^{-/-}$ mice (*Figure 1—figure supplement 1A,B*). Although not detected in endoscopic examination, occasional mucosal erosion (partial loss of mucosa) but not ulceration (complete loss of mucosa) was observed in 30 week old *Il10rb*$^{-/-}$ mice (*Figure 1A*). Furthermore, while the colon length in *Il10rb*$^{-/-}$ mice was not shortened compared to the control mice (*Figure 1—figure supplement 1C*), the colon weights of *Il10rb*$^{-/-}$ mice were significantly increased (p<0.001; *Figure 1—figure supplement 1D*). Notably, growth arrest in *Il10rb*$^{-/-}$ mice was not observed until 8 weeks of age (*Figure 1C*), indicating that growth arrest is an insensitive indicator of intestinal inflammation in this system.

To further characterize the evolution of the inflammatory response in the absence of IL10 signaling, we analyzed gene expression in the colon of mice from birth to 6 months of age. Strikingly,

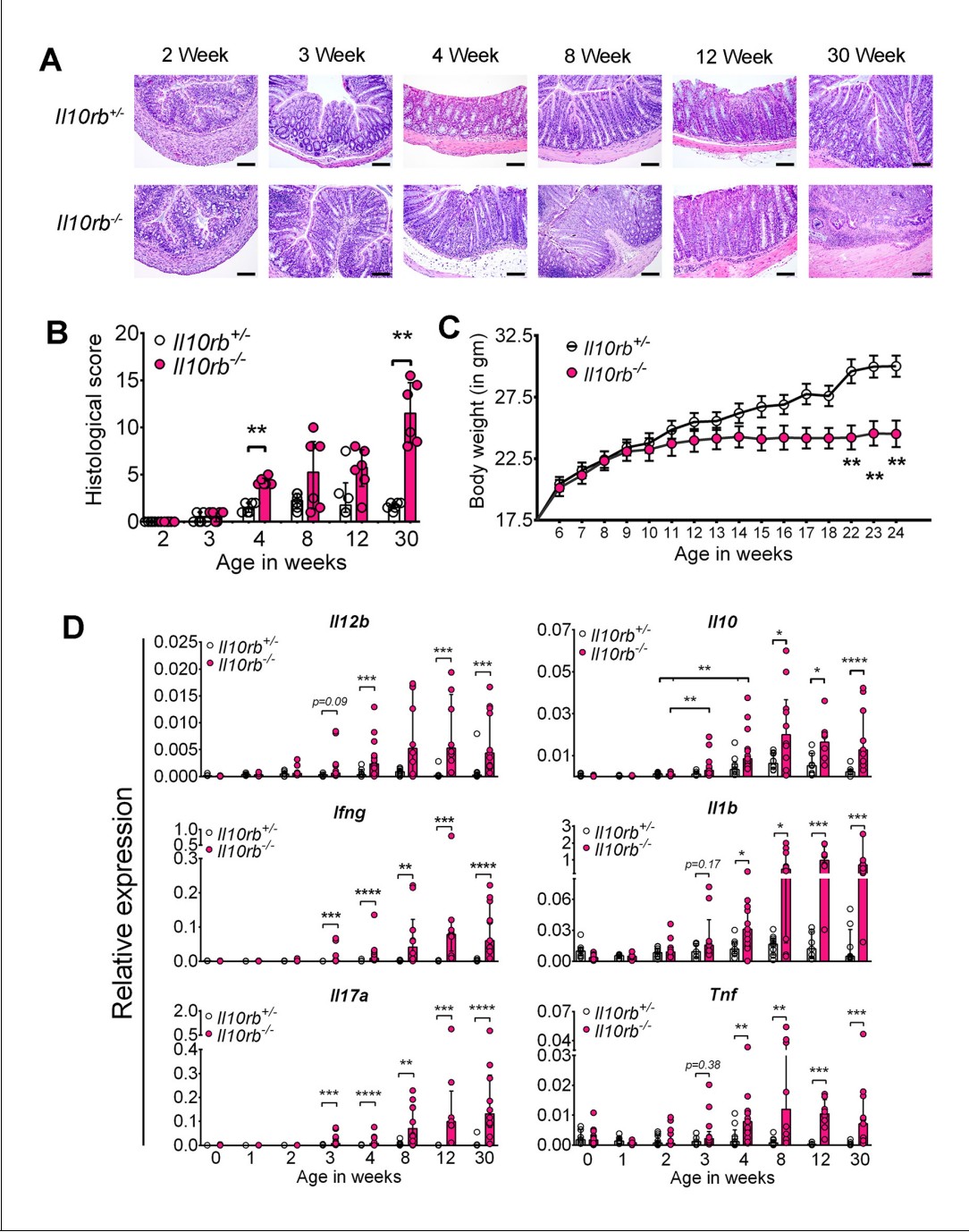

**Figure 1.** Spontaneous colitis develops in *Il10rb⁻ᐟ⁻* mice between 3–4 weeks. (A) Representative histologic images of distal colon from *Il10rb⁻ᐟ⁻* and control mice at indicated age (10X magnification for 8 week and 30 week *Il10rb⁻ᐟ⁻*, scale = 200 µm; and 20X for all other samples, scale = 100 µm). (B) Histologic scores (0–24) from *Il10rb⁻ᐟ⁻* and control mice at indicated age (n = 6 mice/group). Median with interquartile range (IQR) is shown. Significance determined by Mann-Whitney test. (C) Body weight in a cohort of mice from 6 to 24 weeks (n = 10–12). Mean ± SEM of data analyzed by 2-way ANOVA followed by Bonferroni's Multiple Comparison tests is shown; (D) Colonic gene expression assessed by qRT-PCR (n = 5–15 mice in each group). Median with IQR is shown. Significance determined by Mann-Whitney test. *p<0.05, **p<0.01, ***p<0.001, ****p<0.0001. Additional data file (*Figure 1—figure supplement 1*) showing severe colitis in 6 month old *Il10rb⁻ᐟ⁻* mice is provided.

The following figure supplement is available for figure 1:

**Figure supplement 1.** Severe colitis in 6 month old *Il10rb⁻ᐟ⁻* mice.

during the third and fourth week of life, we observed a dramatic increase in proinflammatory cytokine expression including *Il12b, Ifng, Il17a, Il1b,* and *Tnf* within the colon of *Il10rb*[-/-] mice but not in littermate controls (*Figure 1D*). Interestingly, concomitant enhanced expression of *Il10* also started at 3 weeks in both *Il10rb*[-/-] and control mice, suggesting that the induction of IL10 in control mice is itself necessary to curtail a proinflammatory stimulus that begins in the third week of life.

## Innate inflammatory cells accumulate in the colon of infant *Il10rb*[-/-] mice

To examine the cellular basis for inflammatory responses observed within the colon of infant *Il10rb*[-/-] mice, we performed extensive immunophenotyping of colonic LP cells isolated from infant mice at 2 and 3 weeks of age. As shown in *Figure 2A–B*, we noted striking differences in the LP populations of MΦs both between the second and third week of life, and between genotypes. We found that colonic CD11b[+]CD64[+] MΦs that exhibited cell surface expression of Ly6C (Ly6C[+]MHCII[-], and Ly6C[+]-MHCII[+], referred to as P1 and P2 MΦs, respectively) expanded to a much greater degree in *Il10rb*[-/-] mice than in control mice between the second and third week of life, with a concomitant decrease in the percentage of Ly6C[-]MHCII[+] (referred to as population P3/P4 MΦs, gating strategy shown in *Figure 2—figure supplement 1*) (*Figure 2A,B*). Given that the P1 and P2 MΦ populations are thought to represent recently recruited monocytes, while the P3/P4 MΦ population is thought to represent resident IL10-expressing MΦs (*Bain et al., 2014, 2013; Tamoutounour et al., 2012*), these results suggest a marked elevation in newly recruited monocytes within the LP of 3 week old *Il10rb*[-/-] mice. Interestingly, while the relative abundance of P1/P2 versus P3/P4 MΦ were distributed reciprocally in 3 week old *Il10rb*[-/-] mice compared to control mice, the total cell numbers of all MΦ subsets (P1-P4) in the colons of *Il10rb*[-/-] mice were increased compared to control mice (*Figure 2B*), consistent with the hypothesis that the recruitment of monocytes and conversion to MΦs is enhanced within the colonic LP of 3 week old mice lacking IL10Rβ.

Within lymphocyte subsets (Th1, Th17, Treg, ILC3s), significant differences between infant *Il10rb*[-/-] and control mice were only detected for the proportions of FoxP3[+] Tregs, and IFNγ-producing effector T cells; although the magnitude of this difference was relatively small (*Figure 2—figure supplement 2A–C,E–G*). Similarly, there were subtle differences in CD11c[+] DC subsets in 3 week old *Il10rb*[-/-] mice compared to the control mice (*Figure 2—figure supplement 2D,H*). Taken together, these results demonstrate significant perturbations in MΦ phenotype but not other innate or adaptive immune cells within the LP of 3 week old *Il10rb*[-/-] mice.

## Lamina propria macrophages of infant *Il10rb*[-/-] mice possess a proinflammatory transcriptional signature

It has previously been suggested that P3/P4 LP MΦs exhibit anti-inflammatory properties, in part based on low expression of proinflammatory mediators including IL6, iNOS (Nos2), and TNF, and their ability to secrete IL10 (*Bain et al., 2013*). Given that we observed marked alterations in the distribution of LP MΦs in *Il10rb*[-/-] mice, we wondered whether P3/P4 MΦs retained their anti-inflammatory properties in the absence of the IL10Rβ. To examine this issue, we performed RNA sequencing (RNA-seq) on FACS-sorted P3/P4 MΦs from 1, 3, and 12-week-old *Il10rb*[-/-] and control mice. We identified a number of proinflammatory genes including *Il1a, Il12a, Il12b, Il23a,* and *Nos2* that were expressed at higher levels in P3/P4 MΦs isolated from 12 week old *Il10rb*[-/-] mice than from 12 week old control mice (*Figure 3A*). Interestingly, while we did not detect any significant difference in expression of these genes in MΦs isolated from 1 week old mice of either genotype, expression of this set of genes appeared higher in P3/P4 LP MΦs isolated from 3 week old *Il10rb*[-/-] mice than from control mice (*Figure 3A*). We further validated the expression of a subset of these genes by qRT-PCR analysis on FACS-sorted P3/P4 MΦs from 3-4 week old *Il10rb*[-/-] and control mice and found similar results (*Figure 3B*). These data suggest that P3/P4 MΦs isolated from *Il10rb*[-/-] mice lose their anti-inflammatory signature by 3 weeks of age.

## Clodronate liposome-mediated depletion of colonic macrophages ameliorates colitis in infant *Il10rb*[-/-] mice

The observation that P3/P4 LP MΦs isolated from *Il10rb*[-/-] mice exhibit a pro-inflammatory profile led us to consider whether MΦs were required for the inflammatory response observed in 3-week-old *Il10rb*[-/-] mice. To test this hypothesis, we employed clodronate liposome treatment (i.p.) in 3–4

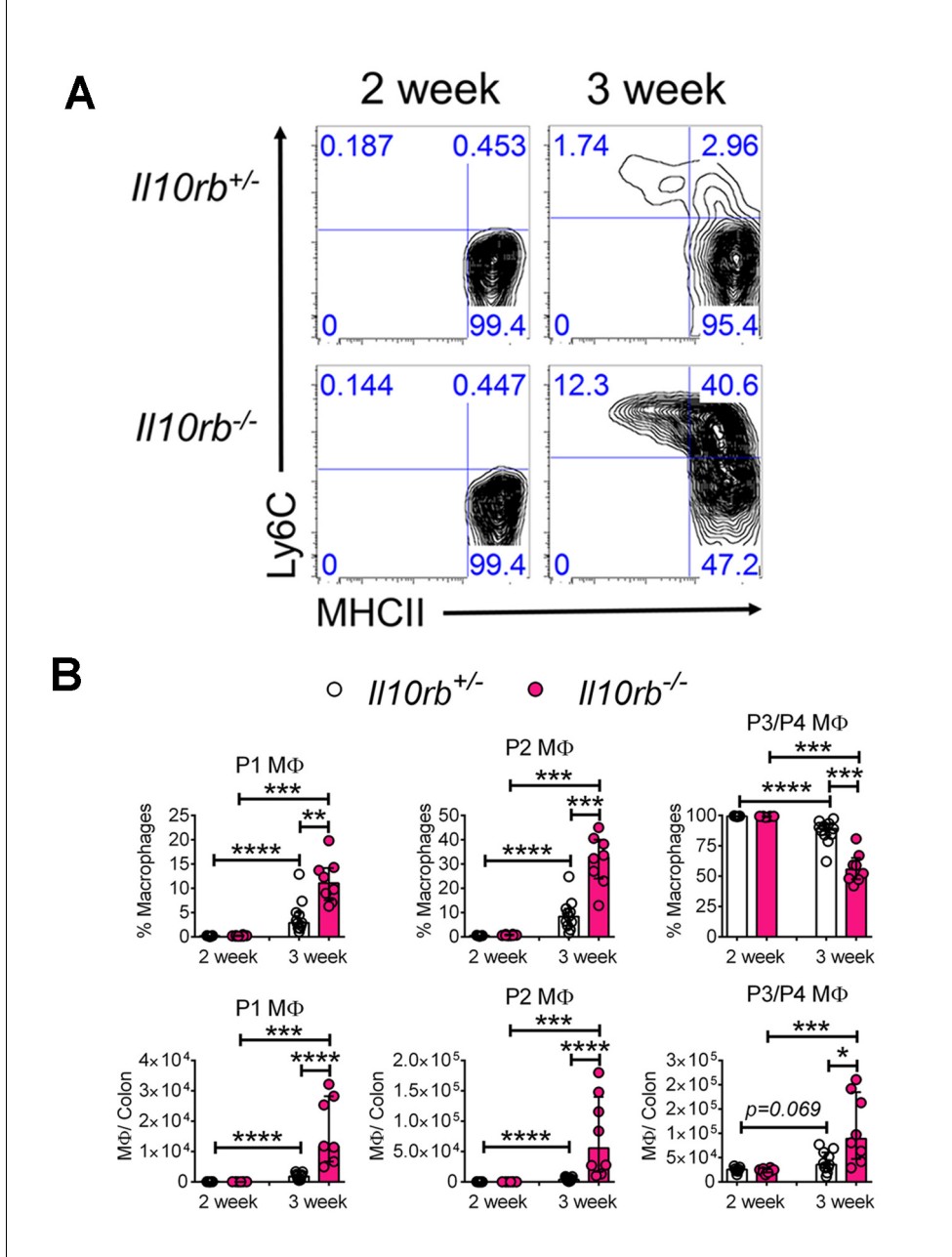

**Figure 2.** Colonic lamina propria macrophage dysfunction in 3–4 week old *Il10rb⁻/⁻* mice. Representative FACS plots of (A) MΦ subsets, gated on CD45⁺CD11b⁺ CD11cᶦⁿᵗCD103⁻CD64⁺ cells in the LP of *Il10rb⁻/⁻* and littermate *Il10rb⁺/⁻* mice at 2 and 3–4 weeks. Gating strategy is shown in the accompanying *Figure 2—figure supplement 1*. Comparison of the frequencies and absolute numbers per colon of (B) LP MΦ subsets is shown for *Il10rb⁻/⁻* vs control *Il10rb⁺/⁻* mice at 2 and 3–4 weeks of age. Results are pooled from 2 to 3 litters leading to 7–11 mice in each group. *p<0.05, **p<0.01, ***p<0.001, ****p<0.0001, Mann-Whitney U test. Data represents Median with IQR. Additional data file (*Figure 2—figure supplement 2*) showing minimal alterations in colonic lymphocytes and dendritic cells in 3–4 week old *Il10rb⁻/⁻* mice is provided.

The following figure supplements are available for figure 2:

**Figure supplement 1.** Gating strategy for colonic lamina propria macrophages.

**Figure supplement 2.** Minimal alterations in colonic lymphocytes and dendritic cells occur in the absence of IL10Rβ signaling in 3–4 week old mice.

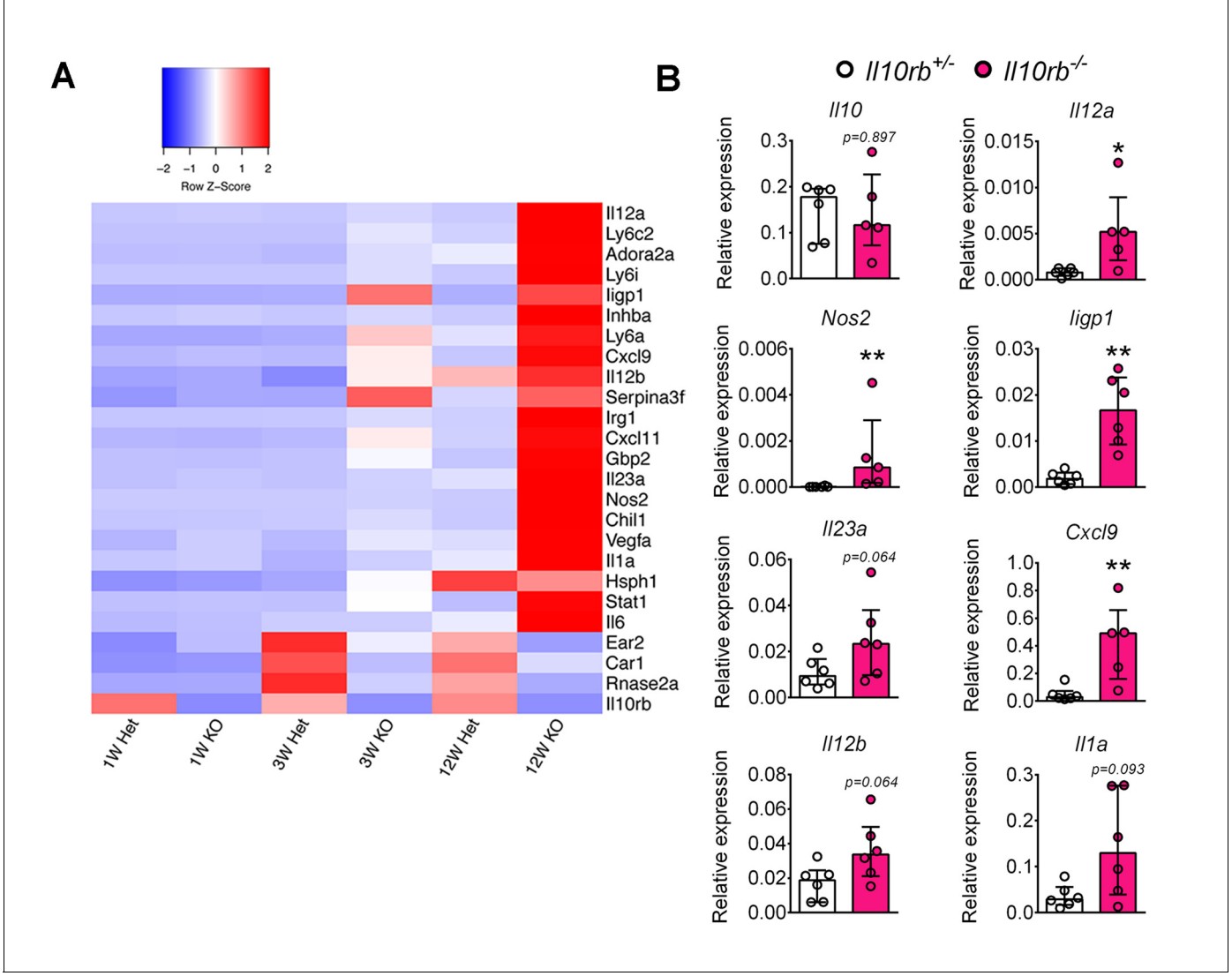

**Figure 3.** Lamina propria macrophages of infant *Il10rb⁻/⁻* mice exhibit proinflammatory transcriptional profile. Colonic LP MΦs (P3/P4; gated on CD45⁺CD11b⁺CD11c^int CD103⁻CD64⁺Ly6C⁻MHCII⁺ cells) were sorted (minimum 10,000 cells), RNA was prepared, and RNA-seq was performed. (A) Heatmap showing relative expression intensity of >3 fold differentially expressed genes in the LP MΦs between *Il10rb⁻/⁻* (KO) and *Il10rb⁺/⁻* (Het) mice. Genes were identified based on the differential expression in LP MΦs between 3 week old *Il10rb⁻/⁻* and control mice. Each sorting time-point represents one mouse at 3 and 12 weeks, whereas pooled colons from 4 to 5 mice for 1 week data points were used. (B) Validation of RNA-seq data by qRT-PCR for selected genes and *Il10* from LP P3/P4 MΦs sorted from *Il10rb⁻/⁻* and littermate *Il10rb⁺/⁻* mice at 3–4 weeks of age. *p<0.05, **p<0.01, n = 6 mice in each group. Median with IQR is shown. Additional data files showing codes used for 'R' scripts (*Figure 3—source data 1* and *Figure 3—source data 2*), and the normalized gene expression values for *Figure 3A* (*Figure 3—source data 3*) are provided.

The following source data is available for figure 3:

**Source data 1.** R-script used to generate differential gene expression data shown in *Figure 3A*.
**Source data 2.** R-script used to generate heat map shown in *Figure 3A*.
**Source data 3.** Normalized gene expression values used to construct heat map shown in *Figure 3A*.

week old *Il10rb*[-/-] mice, a treatment that has previously been shown to deplete LP MΦs and inhibit the development of intestinal inflammation in *Helicobacter bilis* infected *Rag2*[-/-] mice (*Muthupalani et al., 2012*). Clodronate-containing liposomes significantly inhibited the histological signs of colitis in 3–4 week old *Il10rb*[-/-] mice compared to those treated with control liposomes (*Figure 4A,B*). Immunophenotypic analyses revealed that clodronate treatment significantly reduced the abundance of both P2 and P3/P4 MΦs in the colonic LP of *Il10rb*[-/-] mice (*Figure 4C,D*), as well as the percentages and absolute numbers of CD4[+] T cells that produced IL17A and IFNγ (*Figure 4C,E*). In addition, clodronate treatment led to significantly abrogated expression of genes encoding for proinflammatory effector cytokines including *Il12b*, *Ifng*, *Il17a*, and *Il1b* (*Figure 4F*) in the colons of *Il10rb*[-/-] mice. Taken together, these data suggest that the LP MΦs are important effectors of colonic inflammatory response in the absence of IL10R signaling in infant mice.

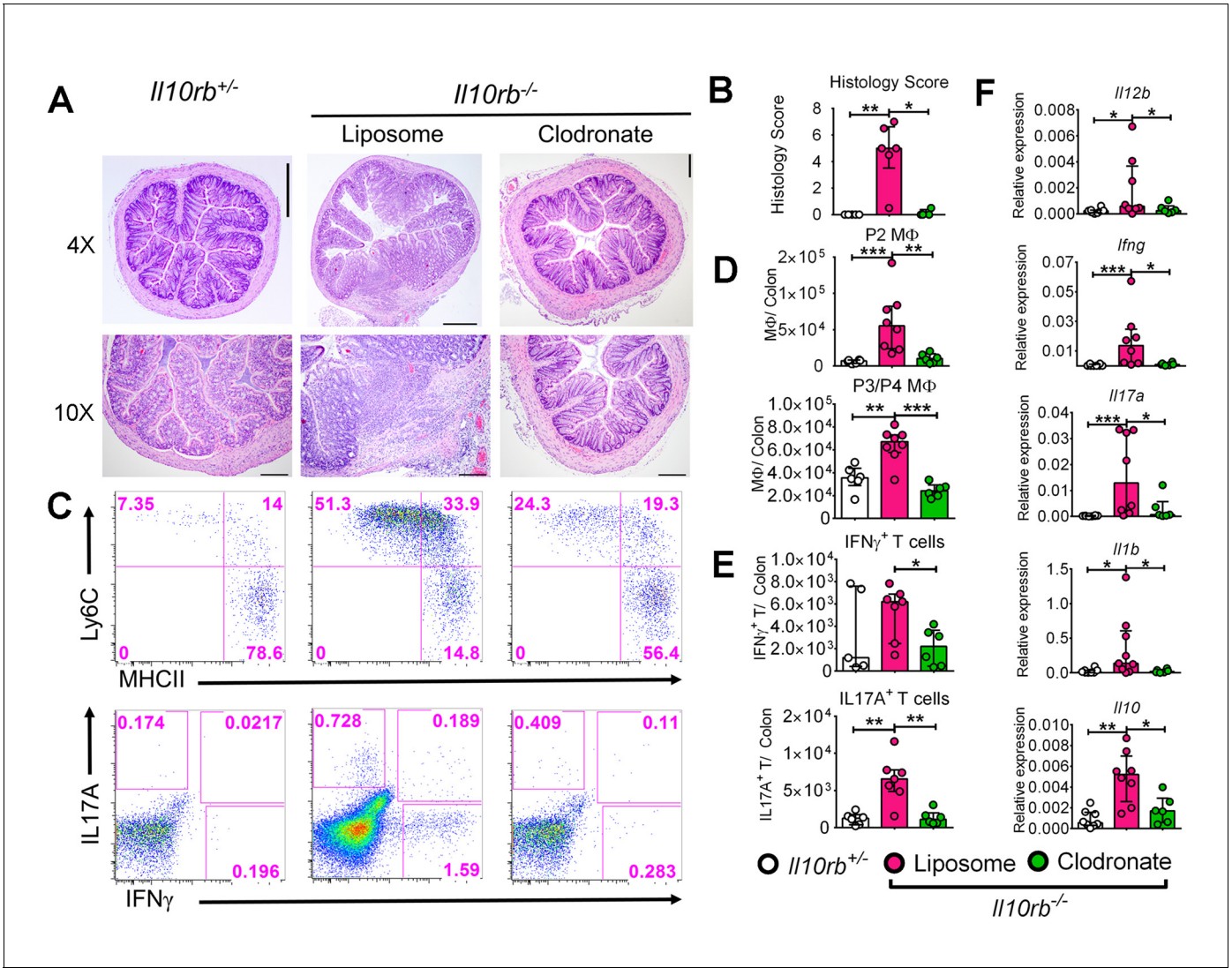

**Figure 4.** Liposomal clodronate-mediated depletion of colonic macrophages ameliorates colitis in infant *Il10rb*[-/-] mice. Three week old *Il10rb*[-/-] mice were injected intraperitoneally (i.p.) with liposomal control or liposomal-clodronate for three times within a week. (A) Representative histologic images of distal colons (4X and 10X magnification). Scale = 500 μm (4X) and 200 μm (10X). (B) Histological scores from 4 to 6 mice in each group. (C) Representative FACS plots of colonic LP MΦ subsets (upper panel) gated on CD45[+]CD11b[+]CD11c[int]CD103[-]CD64[+] cells, and (lower panel) CD4[+] T effector subsets following PMA and ionomycin stimulation for 4.5 hr, gated on CD45[+]CD3e[+]CD4[+] T cells. Comparison of the total numbers per colon of (D) LP MΦs, and (E) CD4[+] T effector cells. (F) Gene expression analysis by qRT-PCR from the colons of mice indicated above. *p<0.05, **p<0.01, ***p<0.001, Mann-Whitney *U* test. Data shows Median with IQR of 2 pooled experiments leading to 4–8 mice in each group.

## IL10Rα chain is essential to prevent inflammation in the developing infant colon

The data presented above demonstrates that IL10Rβ prevents the development of intestinal inflammation between the second and third week of life, suggesting a critical role for IL10 signaling during this developmental period. However, IL10Rβ is also a component of other receptor complexes including the receptors for IL22, IL26, and IFNλ (*Donnelly et al., 2004*). Thus, it is possible that the pathology observed in infant mice is secondary to the absence of additional cytokine receptor signaling (specifically IL22R which has an important role in epithelial cell function) rather than the absence of the IL10 receptor itself. To address these issues, we employed *Il10ra*$^{-/-}$ mice in which the IL22R, IL26R, and IFNλR are intact. As *Il10ra*$^{-/-}$ mice were only available on the C57BL/6 background, and do not develop spontaneous colitis in our vivarium (data not shown), *Il10ra*$^{-/-}$ mice were crossed to cytokine deficiency-induced colitis susceptibility (*Cdcs*)1 congeneic mice (C57BL/6) (*Bleich et al., 2010*). This congenic *Cdcs1* strain harbors a region on chromosome originally identified in C3H/HeJ-Bir mice, which increases colitis susceptibility of immunodeficient mice on the C57BL/6 background (*Beckwith et al., 2005*; *Farmer et al., 2001*). While infant *Cdcs1*$^{+/+}$*Il10ra*$^{+/-}$ mice (control) were asymptomatic, 100% of *Cdcs1*$^{+/+}$*Il10ra*$^{-/-}$ mice developed moderate multifocal proliferative granulocytic colitis with crypt abscesses and loss of goblet cells at 3 weeks (*Figure 5A,B*). By 12 weeks, most of the *Cdcs1*$^{+/+}$*Il10ra*$^{-/-}$ mice developed severe diffuse proliferative granulocytic colitis with loss of goblet cells (data not shown). Consistent with data from *Il10rb*$^{-/-}$ mice (*Figure 2A,B*), the 3 week old *Cdcs1*$^{+/+}$*Il10ra*$^{-/-}$ mice also displayed increased numbers of LP MΦ populations and a skewing of the frequency towards the P1/P2 populations (*Figure 5C,D*), and increased IFNγ$^+$ CD4$^+$ effector T cells (data not shown). Analysis of colonic gene expression also demonstrated significantly higher expression of genes for proinflammatory effectors such as *Il12b, Ifng, Il17a, Il1b*, and *Tnf* in the 3 week old *Cdcs1*$^{+/+}$*Il10ra*$^{-/-}$ mice than observed in 3 week old control *Cdcs1*$^{+/+}$*Il10ra*$^{+/-}$ mice (*Figure 5E*). These data strongly support the hypothesis that in this model IL10R signaling is necessary to prevent the onset of colitis between the second and third weeks of life, and has critical functions that are independent of the related receptors for IL22, IL26, and IFNλ. However, it is important to note that as we did not directly compare the phenotype of mice lacking IL10Rα and IL10Rβ, there may be difference in the phenotype of mice lacking these two components based on differing functions of these receptor components themselves, or conversely due to differences in genetic background or resident microbiota in these two strains.

## IL10Rα signaling in macrophages is required to prevent the development of colitis in infant mice

The data presented thus far is consistent with the notion that absence of IL10R in infant mice leads to alterations in colonic MΦ phenotype that promotes colitis. However, C57BL/6 mice that specifically lack IL10Rα in MΦs (e.g. *Il10ra*$^{fl/fl}$*Lyz2*$^{Cre}$ mice) either do not reliably develop colitis (*Li et al., 2015*) or do so with a prolonged latency (e.g. *Il10ra*$^{fl/fl}$*Cx3cr1*$^{Cre}$ mice) (*Zigmond et al., 2014*). The accelerated colitic phenotype we observed in the *Cdcs1*$^{+/+}$*Il10ra*$^{-/-}$ mice presented the opportunity to re-address this issue. Therefore, we crossed *Il10ra*$^{fl/fl}$*Lyz2*$^{Cre}$ mice with the *Cdcs1*$^{+/+}$ strain to generate both *Cdcs1*$^{+/+}$*Il10ra*$^{fl/fl}$*Lyz2*$^{Cre}$ mice and control *Cdcs1*$^{+/+}$*Il10ra*$^{fl/f}$ mice. At 6 weeks of age, there were gross signs of colitis including thickened colons and poorly formed stool pellets in *Cdcs1*$^{+/+}$*Il10ra*$^{fl/fl}$*Lyz2*$^{Cre}$ mice that were absent within the colons of *Cdcs1*$^{+/+}$*Il10ra*$^{fl/f}$ mice (*Figure 6A*). Histologic analysis revealed that the distal colons of 100% of 3–4 week old *Cdcs1*$^{+/+}$*Il10ra*$^{fl/fl}$*Lyz2*$^{Cre}$ mice exhibited mild-moderate colonic inflammation that was further augmented by 6 weeks with significantly increased inflammatory infiltrates, hyperplasia, and loss of goblet cells, whereas the colons of *Cdcs1*$^{+/+}$*Il10ra*$^{fl/f}$ mice were histologically normal (*Figure 6B,C*). Consistent with the presence of inflammation, we observed increased expression of inflammatory genes including *Ifng, Il17a, Il1b*, and *Tnf* within the colons of *Cdcs1*$^{+/+}$*Il10ra*$^{fl/fl}$*Lyz2*$^{Cre}$ mice compared to *Cdcs1*$^{+/+}$*Il10ra*$^{fl/fl}$ mice (*Figure 6D*). Further, the absence of IL10Rα specifically on MΦs led to significantly dysregulated LP pro- (P1/P2) and anti-inflammatory P3/P4 MΦ frequency and numbers (*Figure 6E–F*). However, we did not detect statistically significant changes in the frequencies or numbers of LP IFNγ$^+$, IL17A$^+$, or IFNγ$^+$IL17A$^+$ CD4$^+$ T cells, or FoxP3$^+$ Tregs in 3–4 week old *Cdcs1*$^{+/+}$*Il10ra*$^{fl/fl}$*Lyz2*$^{Cre}$ mice compared to *Cdcs1*$^{+/+}$*Il10ra*$^{fl/fl}$ mice (*Figure 6—figure supplement 1*). Together, these data confirm that IL10Rα signaling on MΦs is necessary to protect from intestinal inflammation

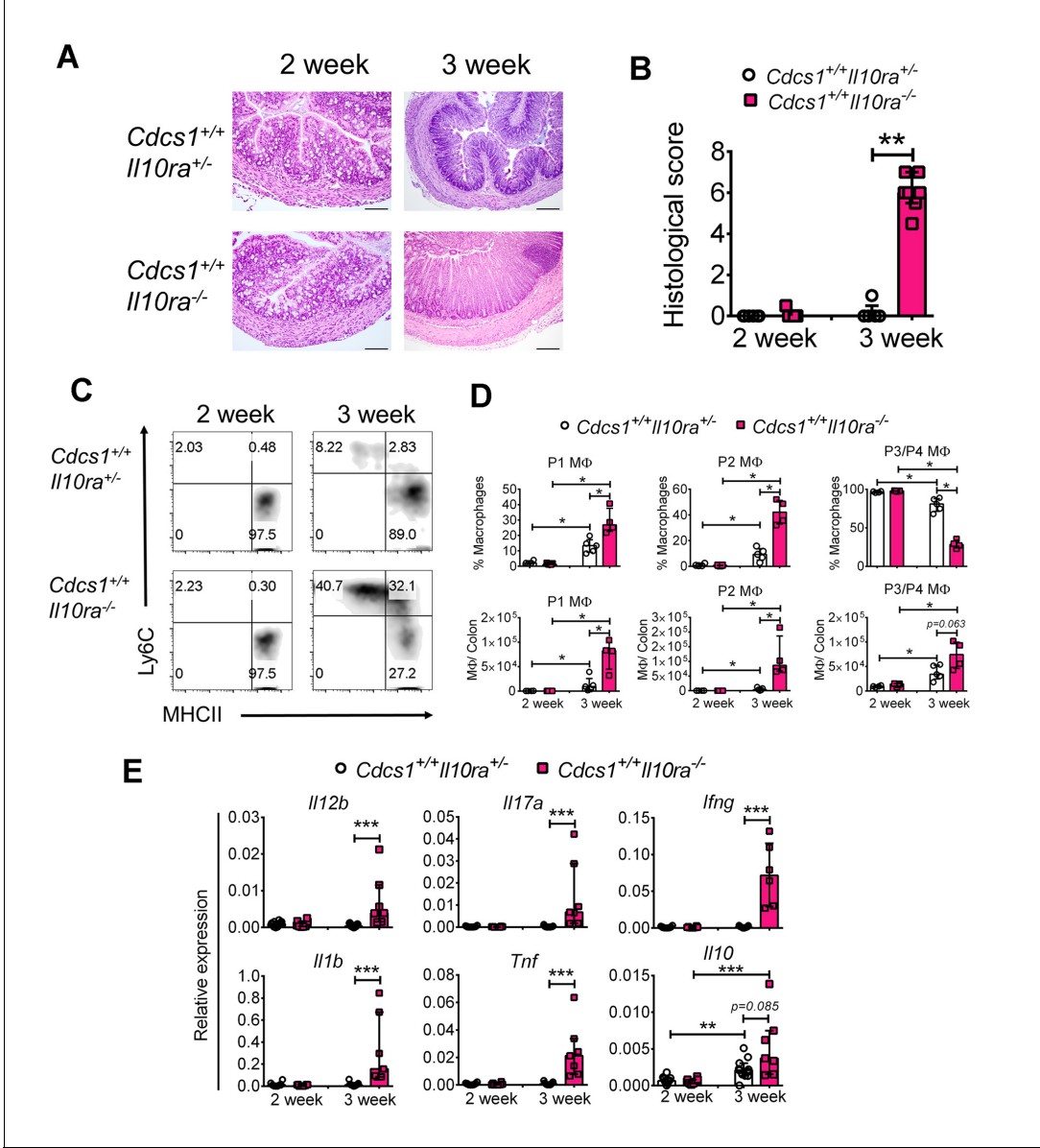

**Figure 5.** IL10Rα is necessary to prevent intestinal inflammation in infant mice. *Il10ra^-/-* mice (C57BL/6) were crossed with congenic *Cdcs1* mice and the development of spontaneous colitis was investigated. (A) Representative histologic images from the colon of *Cdcs1^+/+Il10ra^-/-* and littermate *Cdcs1^+/+Il10ra^+/-* control mice at indicated ages (bar represents 100 μm for 2 weeks and 200 μm for 3 weeks). (B) Histologic scores from indicated mice; (C) Representative FACS plots of colonic LP MΦ populations, gated on CD45^+ CD11b^+CD11c^intCD103^-CD64^+ cells. (D) Comparison of the frequencies and numbers of LP MΦ subsets in the colons of mice described. (E) Summary of gene expression analysis by qRT-PCR from the colons of mice indicated above. *p<0.05, **p<0.01, ***p<0.001, Mann-Whitney *U* test, Median with IQR is shown. Data were pooled from 2 to 3 litters resulting in a total of 4–11 mice in each group.

during murine infant development and that IL10Rα modulates the phenotype of LP MΦs in a cell-intrinsic fashion.

## Intestinal inflammation in infant *Il10rb^-/-* mice is microbiota-dependent

The data presented above demonstrate that the absence of IL10R in several different models leads to the development of intestinal inflammation and LP MΦ dysfunction between the second and third weeks of life. This suggests that during this developmental window a pathway is activated with the potential to induce intestinal inflammation in the absence of physiological IL10R signaling in MΦs.

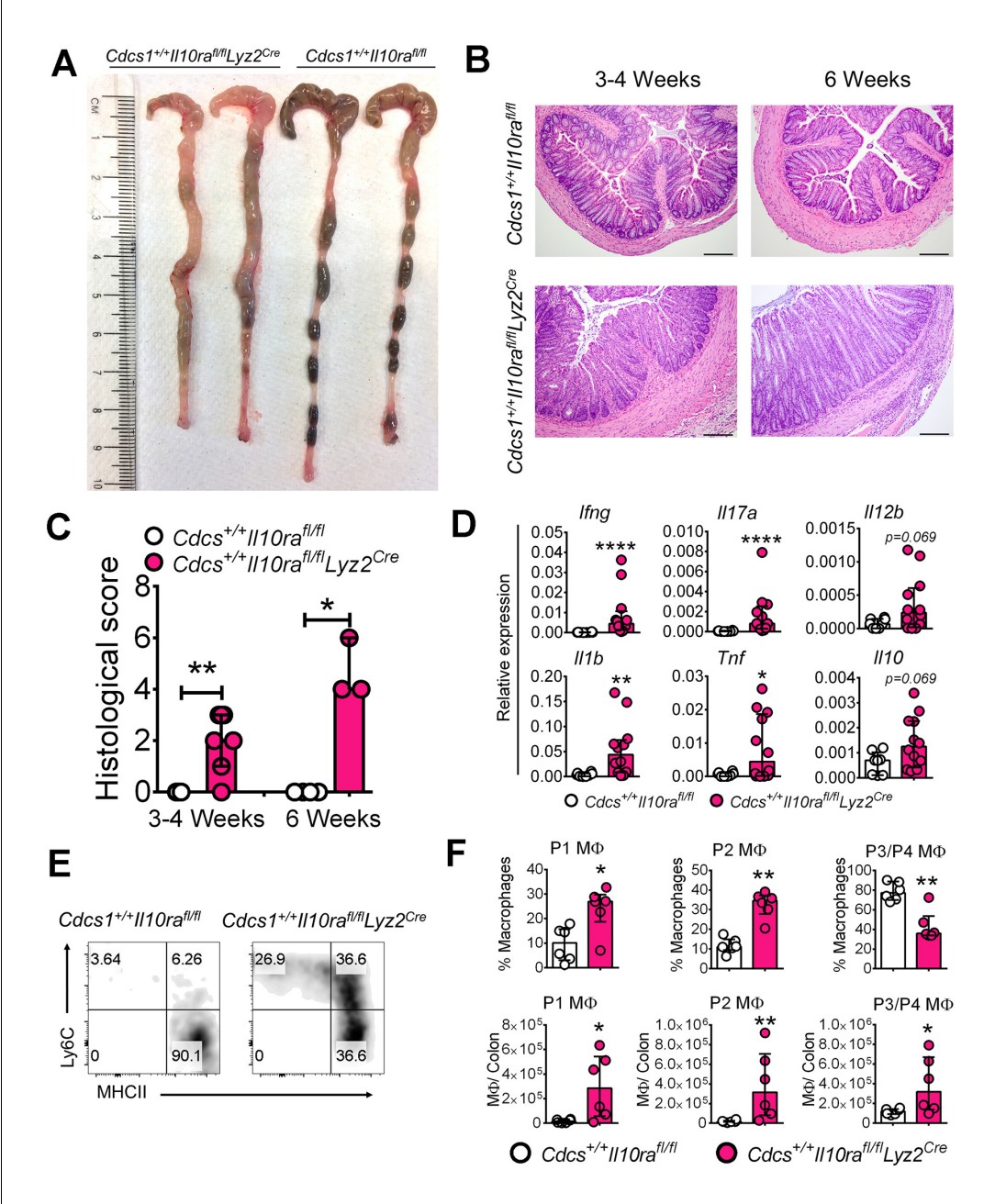

**Figure 6.** Macrophage-specific IL10Rα is necessary to prevent intestinal inflammation in infant mice. (A) Representative gross macroscopic images of colon from 6 week old mice. (B) Representative histologic images of distal colons of *Cdcs1^{+/+}Il10ra^{fl/fl}Lyz2^{Cre}* and littermate *Cdcs1^{+/+}Il10ra^{fl/fl}* control mice (bar = 200 μm). (C) Histologic scores from indicated mice. (D) Colonic gene expression analysis by qRT-PCR. (E) Representative FACS plots; and (F) frequencies and numbers of colonic LP MΦ populations in the colons of 3–4 week old mice. *p<0.05, **p<0.01, ****p<0.0001, Mann-Whitney *U* test, Median with IQR is shown. Each data point indicates individual mouse, data were pooled from 2 to 3 litters resulting in a total of 4–12 mice in each group; whereas a representative of 2 litters is shown for 6 week old mice in A-C. Additional data file (*Figure 6—figure supplement 1*) showing the frequency and numbers of colonic CD4^+ lymphocytes in 3–4 week old *Cdcs1^{+/+}Il10ra^{fl/fl} Lyz2^{Cre}* and littermate *Cdcs1^{+/+}Il10ra^{fl/fl}* mice is provided.

The following figure supplement is available for figure 6:

**Figure supplement 1.** Immunophenotyping of 3–4 week old *Cdcs1^{+/+}Il10ra^{fl/fl} Lyz2^{Cre}* and littermate *Cdcs1^{+/+}Il10ra^{fl/fl}* control mice.

One significant change that occurs during this period is the transition from breast milk to solid food, and the rapid diversification in the intestinal microbiome that ensues (*Schloss et al., 2012*). It has previously been shown that antibiotics can inhibit colitis that develops in infant IL10Rβ-deficient mice expressing a dominant negative (dn) TGFβRII in T cells (*Kang et al., 2008*), and further that Gram-positive bacteria promote monocyte/ MΦ recruitment into the colonic LP during DSS colitis (*Nakanishi et al., 2015*). However, the role of the intestinal microbiota in promoting alterations in MΦ phenotype and inflammation in mice solely lacking the IL10 receptor has not been determined. To assess this, we added a cocktail of broad-spectrum antibiotics including metronidazole, vancomycin, ampicillin, and neomycin (termed Abx), or selective antibiotics that preferentially target Gram-negative organisms (polymyxin B) or Gram-positive organisms (vancomycin) to the drinking water (*Atarashi et al., 2011*) of pregnant dams, which was then continued after pups were born. Antibiotic or control treated pups were weaned at 3 weeks into cages supplemented with the respective antibiotics or regular water, and then euthanized at 4 weeks. As expected, 4 week old *Il10rb*$^{-/-}$ mice maintained on regular water developed moderate colitis (*Figure 7A*). Notably, 4-week-old mice maintained on the antibiotic cocktail or polymyxin B showed a significant reduction in microscopic signs of colitis (*Figure 7A,B*). In contrast, while we noted that the median inflammatory score was lower in vancomycin-exposed *Il10rb*$^{-/-}$ mice than in controls, this difference did not reach statistical significance (*Figure 7A,B*).

With regards to the influence of microbiota on MΦ phenotype, analysis of colonic LP MΦs showed that the Abx cocktail or polymyxin B significantly reduced the expansion of P1-P2 subsets while simultaneously restoring normal proportions of the P3/P4 subsets of MΦs in *Il10rb*$^{-/-}$ mice (*Figure 7C,D*). Consistent with histopathological grading (*Figure 7B*), however, the distribution of pro- and anti-inflammatory LP MΦ distribution in *Il10rb*$^{-/-}$ mice treated with vancomycin was not significantly changed compared to regular water controls (*Figure 7C,D*). Furthermore, the treatment of *Il10rb*$^{-/-}$ mice with Abx cocktail prevented the colonic expression of *Il10* as well as major pro-inflammatory effector genes including *Il12b*, *Il17a*, and *Ifng* compared to *Il10rb*$^{-/-}$ mice that did not receive antibiotics (*Figure 7E*). While treatment with polymyxin B also significantly reduced the expression of these genes except *Il17a*, differences in gene expression levels following exposure to vancomycin did not achieve statistical significance (*Figure 7E*). Collectively, these data indicate that microbiota are a key factor that regulate MΦ function and promote colonic inflammation during early postnatal development in *Il10rb*$^{-/-}$ mice. In addition, the specificity of antibiotics indicate that Gram-negative bacteria likely represent key effectors of the inflammatory response observed in *Il10rb*$^{-/-}$ mice during this developmental window. Using the sample sizes reported here (9–11 mice/group) differences in inflammatory pathology between untreated *Il10rb*$^{-/-}$ mice and those treated with vancomycin did not reach statistical significance. However, we did note a substantial amount of variability in these experiments, suggesting that significance might be achieved with larger sample sizes. Therefore, at this time, we cannot unambiguously assign a role for Gram-positive bacteria in driving colonic inflammation in this model.

## Microbiome expands in the developing intestine independent of IL10Rβ signaling

The data above indicates that the IL10R protects infant mice from microbiota-driven colonic inflammation. There are at least two possible explanations for this phenomenon. One possibility is that IL10R is necessary on MΦs to prevent excessive microbially-induced inflammation. The second is that the presence of IL10R on MΦs prevents the acquisition of inflammatory microbiota and the development of a dysbiotic state. To evaluate these possibilities, we collected fecal pellets from a cohort of *Il10rb*$^{-/-}$ and littermate *Il10rb*$^{+/-}$ controls co-housed in an SPF facility every other day from day 17 until day 28 and weekly thereafter up to 14 weeks. We then isolated DNA from these fecal samples and performed pyrosequencing of 16S ribosomal RNA gene amplicons. Assessment of microbiota diversity by the Chao1 index showed that in both groups of mice (*Il10rb*$^{-/-}$ and *Il10rb*$^{+/-}$), microbial community diversity increased rapidly from day 17 to day 35 as expected during weaning, and then plateaued (*Figure 8A*). However, no overall differences in microbial community diversity were observed between genotypes over the time series (p>0.05). Similarly, Shannon entropy analysis (*Shannon, 1997*) did not show any significant differences in microbial diversity between genotypes (data not shown). In addition, weighted Unifrac-based comparison of the fecal microbial communities (a measure of beta diversity) in *Il10rb*$^{-/-}$ and *Il10rb*$^{+/-}$ mice demonstrated similar community

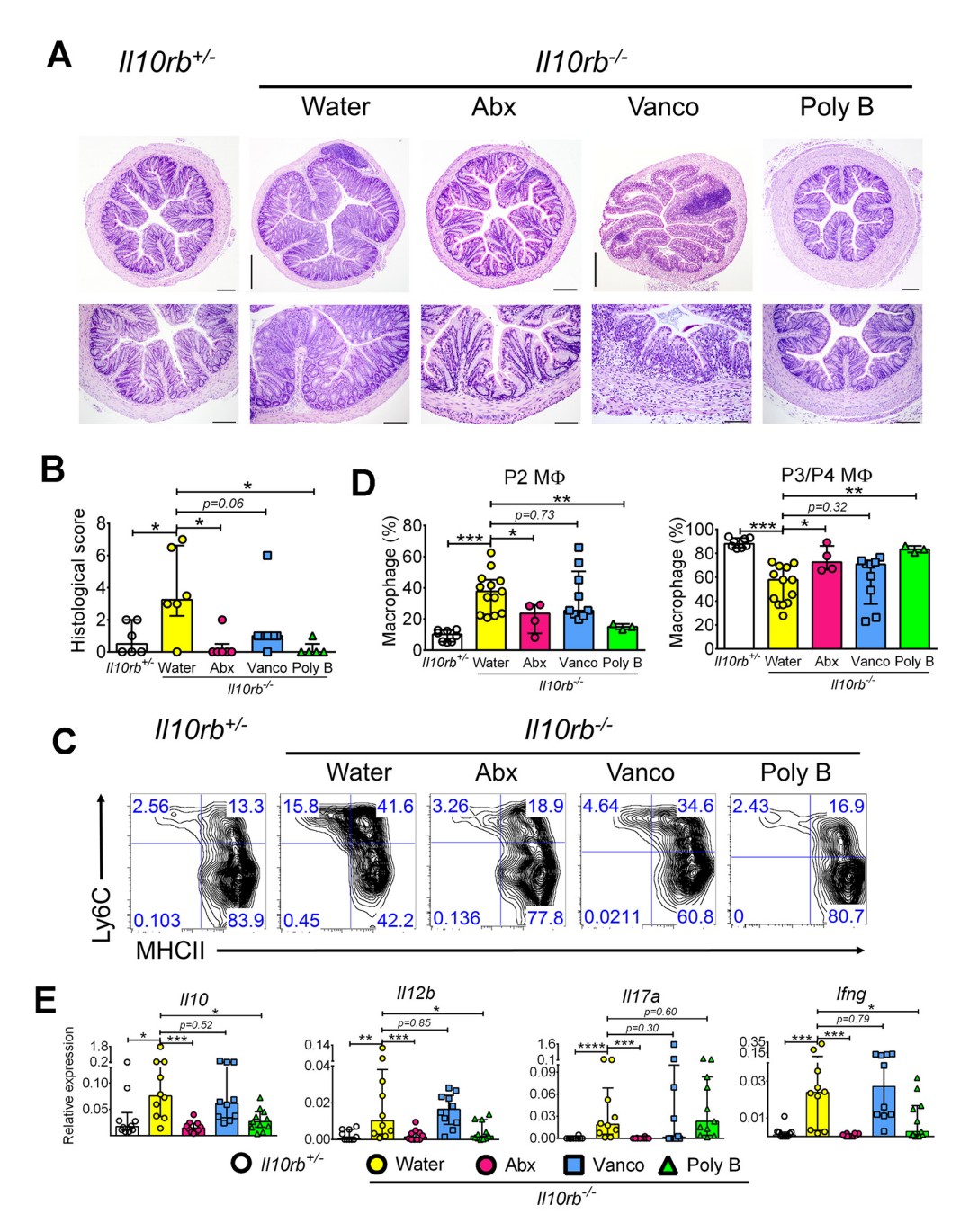

**Figure 7.** Antibiotics exposure prevents colitis in infant *Il10rb⁻/⁻* mice. (A) Representative histological images (4X magnification in upper panel, scale = 500 μm; 10X in lower panel, scale = 200 μm for *Il10rb⁺/⁻*, *Il10rb⁻/⁻* Water, and Poly B; and 20X in lower panel, scale = 100 μm for Abx and Vanco) of distal colons from 4 week old *Il10rb⁻/⁻* or control mice exposed to regular water or indicated antibiotics. (B) Summary of histological scores from 4 to 6 mice. (C) Representative FACS plots of colonic LP MΦ populations, (D) Frequencies of LP MΦ subsets in the colons of mice described in C. (E) Gene expression analysis by qRT-PCR from the colons of mice indicated above. *p<0.05, **p<0.01, ***p<0.001, ****p<0.0001, Mann-Whitney *U* test. Data were pooled from mice (n = 3–11) obtained from 2 to 3 independent experiments. Poly B, polymyxin B; Vanco, vancomycin.

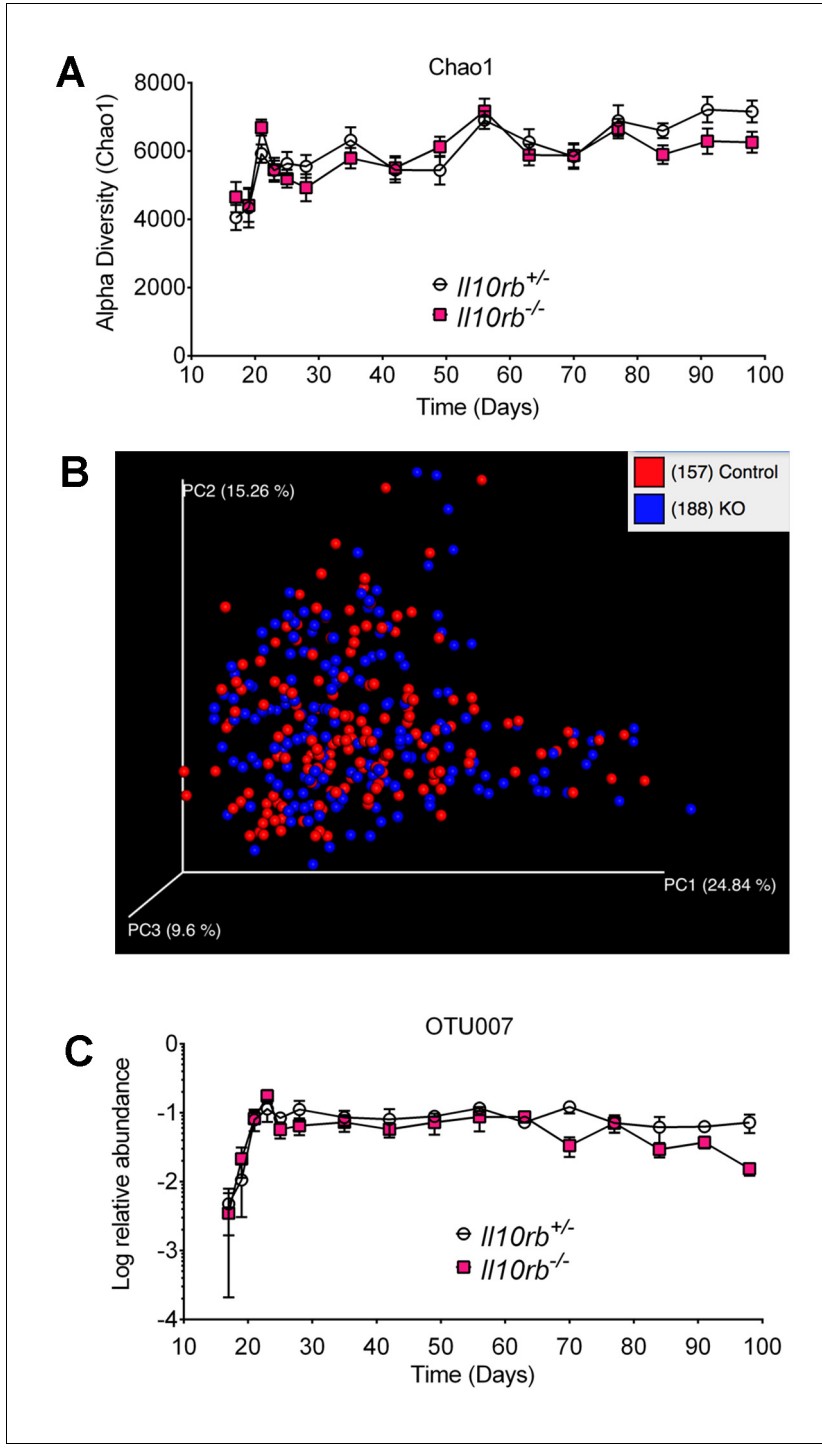

**Figure 8.** IL10Rβ does not regulate fecal microbiome in developing mice. Fecal pellets were collected aseptically every other day until day 28 and then weekly until day 98 from *Il10rb*[-/-] and littermate *Il10rb*[+/-] mice co-housed under SPF conditions. Composition of the microbiota was interrogated via 16S rRNA amplicon pyrosequencing. (**A**) Comparison of alpha diversity (Chao1), and (**B**) Beta diversity (weighted Unifrac). (**C**) Relative abundance (log) of operational taxonomic unit (OTU007, mapping most closely to *Helicobacter ganmani*). Data in (**A**) and (**C**) shows Mean ± SEM of 10–12 mice per genotype, analyzed by 2-way ANOVA followed by Tukey's Multiple Comparison tests. No significant differences between genotypes were found at any point of time ($p>0.05$). The raw values of (**A**) alpha diversity (Chao1), and (**C**) quantification of OTU007 are provided in additional data file (***Figure 8— source data 1***).

*Figure 8 continued on next page*

*Figure 8 continued*

The following source data is available for figure 8:

**Source data 1.** Raw values of alpha diversity (Chao1), and quantification of OTU007 shown in *Figure 8A and C*, respectively.

composition regardless of genotype (*Figure 8B*). Detailed analysis of differential abundances of operational taxonomic units (OTUs), with 393 OTUs present after filtering through our bioinformatics pipeline, revealed few significant differences between $Il10rb^{-/-}$ and $Il10rb^{+/-}$ mice over the time series (data not shown).

Enterohepatic *Helicobacter* species have been shown to drive intestinal inflammation in several immunodeficient murine strains (*Fox et al., 2011*; *Kullberg et al., 1998*). We found that both $Il10rb^{-/-}$ and $Il10rb^{+/-}$ mice acquired *Helicobacter sp.* (Otu007, mapping most closely to *H. ganmani*) at day 17 with the relative abundance increasing up to 4 weeks, which remained stable thereafter (*Figure 8C*). However, the relative abundance of *H. ganmani* remained comparable between both $Il10rb^{-/-}$ and $Il10rb^{+/-}$ mice throughout the time series (p>0.05). These findings suggest that IL10R$\beta$ does not directly regulate microbiota composition, and any ecosystem alterations in $Il10rb^{-/-}$ mice are likely secondary to inflammation induced by the pathogen. Thus, it appears unlikely that IL10R is necessary to control dysbiosis, or that dysbiosis is necessary to promote colitis in infant IL10R-deficient mice.

## Discussion

IBD including CD and UC typically present during adolescence and early adulthood. However, rare patients with causative single gene defects can present in infancy (age <2), or at an age less than 6 years (VEOIBD) (*Uhlig et al., 2014*). A prime example of this is IL10R deficiency which often presents in the first 2 years of life (*Shouval et al., 2014a*; *Glocker et al., 2011*). However, because commonly used mouse models lacking IL10R signaling do not exhibit overt signs of colitis until adulthood, there is a paucity of information regarding requirements for IL10R signaling during infancy. Given the rapid developmental changes in intestinal physiology, immune system responsiveness, diet, and the microbiome that occur during the first weeks of life (*Arrieta et al., 2014*; *Pantoja-Feliciano et al., 2013*), characterizing the function of IL10R signaling during this period could reveal fundamental regulatory processes that are essential to preventing IBD in infancy, and may aid our understanding of the pathological mechanisms driving IBD in general. To this end, we have provided a detailed analysis of the function of IL10R signaling during infant development using a series of genetically modified mice.

Here we demonstrate that although there are no signs of inflammation within the intestine of IL10R-deficient mice less than 14 days of age, IL10R is required to prevent an intestinal inflammatory response that emerges during the third week of life. Consistent with previous work suggesting that IL10R signaling in MΦs prevents excessive lower bowel inflammation in various models (*Li et al., 2015*; *Shouval et al., 2014b*; *Zigmond et al., 2014*), the onset of intestinal inflammation in 3 week old $Il10rb^{-/-}$ mice was associated with increased numbers of LP MΦs, as well as an inversion in the ratio of newly recruited P1/P2 MΦ populations and resident P3/P4 MΦs. Further, while resident P3/P4 MΦs typically exhibit an anti-inflammatory phenotype, RNA-seq analyses indicated that the anti-inflammatory phenotype of IL10R-deficient LP MΦs was compromised at the onset of inflammation, suggesting that altered MΦ phenotype had a causative role in initiating the inflammatory process. In contrast to a clear alteration in LP MΦ phenotype between 2 and 3 week old IL10R-deficient mice, significant alterations in effector T cell populations were not consistently observed. Consistent with a key role for MΦs, we found that clodronate-mediated depletion of MΦs inhibited colitis, and that the lack of IL10Rα specifically on MΦs led to the development of aggressive colitis in 3–4 week old C57BL/6 *Cdcs1* congenic mice. Interestingly, while treatment with antibiotics suppressed inflammation in 3–4 week old mice lacking the IL10R, we found no evidence that IL10R regulated microbiota composition during infancy. These results strongly support a model in which IL10R signaling is required on MΦs to prevent a microbiota-driven inflammatory response that develops during the third week of infant life. Understanding the factors that drive this response, and delineating how

IL10 inhibits this, could uncover essential pathways that regulate the development of intestinal inflammation in infants.

A central question raised by this study is why colonic inflammation in IL10R-deficient mice is first detected in the third week of life. Certainly, one of the most obvious changes in this time frame is the transition from breast milk to solid food, with the concomitant introduction of complex dietary carbohydrates. Multiple aspects of physiology change at the time of weaning (weaning transition), including rapid expansion of the microbiota, alterations in intestinal metabolites (*Arrieta et al., 2014*; *Pantoja-Feliciano et al., 2013*; *Blacher et al., 2017*; *Torow et al., 2017*), and the replacement of yolk-sac derived intestinal MΦs with MΦs derived from circulating monocytes (*Bain et al., 2014*). While hard-wired developmental programs may underlie some aspects of these events, it is conceivable that preventing access to solid food may interfere with expansion of the microbiota observed at the time of weaning, and further that the presence of intestinal microbiota catalyzes the replacement of yolk-sac derived intestinal MΦs with monocyte-derived MΦs (*Bain et al., 2014*). Thus, the weaning transition appears to induce a series of events that remodels intestinal homeostasis, and in the process, activates a pathway whose inflammatory potential is held in check by IL10R signaling.

The role of microbiota in driving intestinal inflammation in immunodeficient strains is well appreciated, and it has been suggested that dysbiosis caused by underlying immune dysregulation may be sufficient to cause colitis in some situations (*Garrett et al., 2007*). While our studies are consistent with previous studies demonstrating that colitis in IL10-deficient mice is attenuated under germ-free conditions (*Sellon et al., 1998*), and that antibiotics can prevent colitis in mice lacking IL10Rβ, or those lacking IL10Rβ and concurrently expressing dnTGFβRII (*Kang et al., 2008*), the lack of dysbiosis in mice deficient in IL10R suggests that IL10 is not regulating the microbiome per se. Therefore, it appears unlikely that a direct effect of IL10R signaling on the microbiome is responsible for the colitis that develops in infant IL10R-deficient mice. In contrast, it was previously reported that IL10-deficient and WT mice exhibit significant differences in microbial diversity and composition (*Maharshak et al., 2013*). Differences between our results and these previous results might be due to differences in the experimental design, endogenous institutional microbiota, and the nature of the animals models used. For instance, in our study IL10R-deficient and control mice were born and co-housed under SPF condition, whereas in the prior study adult germ-free IL10-deficient and WT mice were recolonized with SPF microbiota and then housed separately (*Maharshak et al., 2013*). Nonetheless, the fact that loss in microbial diversity followed the increase in intestinal inflammation in IL10-deficient mice suggested that dysbiosis was a secondary event to inflammation in the IL10-deficient model (*Maharshak et al., 2013*). On the other hand, it is certainly possible that the increase in bacterial diversity that is observed at the weaning transition leads to colonization by microbes with augmented inflammatory potential compared to those present in the immediate postnatal period, such as the potential pathobiont *H. ganmani*, thereby revealing a requirement for IL10R signaling that was not present prior to weaning. Notably, it has been shown that colonization of germ-free IL10-deficient mice with *H. hepaticus* and *Lactobacillus reuteri* but not either species alone, led to severe typhlocolitis in germ-free IL10-deficient mice (*Whary et al., 2011*), indicating that colonization with a Gram-negative pathobiont (*Helicobacter*) in association with other microbes is sufficient to drive intestinal inflammation in susceptible murine strains, and raising the possibility that the expansion of *H. ganmani* in the context of other non-pathogenic flora is responsible for development of infantile colitis in our model. The previous observation that inflammation in *Il10*[-/-] mice, positive for *Helicobacter sp.*, is abrogated in mice specifically lacking the TLR signaling adaptor MyD88 in mononuclear phagocytes (*Hoshi et al., 2012*) indicates the likely importance of direct pathogen recognition, including that of *Helicobacter sp.* and potential other colitis-triggering microbes, for the development of intestinal inflammation in the absence of IL10R signaling.

The increase in microbial diversity observed at the weaning transition also coincides with the initiation of monocyte recruitment into the LP, and replacement of yolk sac-derived LP MΦs with monocyte-derived MΦs (*Bain et al., 2014*). It is unclear whether inflammation in infant *Il10rb*[-/-] mice is driven by resident MΦs or the newly recruited monocyte-derived MΦs. The data from our clodronate experiments demonstrated that a phagocytic cell is responsible for driving inflammation in the infant IL10R-deficient mice but whether this is due to newly recruited monocyte/MΦs is an open question. Interestingly, while blocking monocyte recruitment by using a CCR2-neutralizing antibody (MC21) inhibited DSS-induced colitis (*Zigmond et al., 2012*), in a preliminary experiment we were unable to

detect significant differences in the colonic expression of inflammatory genes including *Il12b*, *Ifng*, *Tnf*, and *Il1b* between isotype and MC21 antibody-treated 3 week old *Il10rb*$^{-/-}$ mice despite markedly decreased infiltration of Ly6C$^{+}$ cells (data not shown). Consistent with this preliminary experiment, it has previously been demonstrated that genetic loss of CCR2 did not reverse colonic inflammation in IL10-deficient mice (*Hoshi et al., 2012*). Notably, monocyte recruitment was not completely blocked in *Il10*$^{-/-}$*Ccr2*$^{-/-}$ mice, suggesting that there might be alternate and potentially redundant mechanisms of monocyte recruitment (*Hoshi et al., 2012*) in these models, and therefore the question of exactly which clodronate-sensitive monocyte/MΦ population drives inflammation in IL10/IL10R deficiency remains open.

Although we did not perform lineage-tracing experiments, by evaluating macrophage phenotype in early infancy we have presumably examined the function of IL10R-deficient MΦs that are of yolk-sac origin. While it has been previously proposed that the function of yolk sac and monocyte-derived LP MΦs are quite similar (*Bain et al., 2014*), we were surprised to find that prior to the weaning transition, the gene expression profiles of LP MΦs of presumed yolk sac origin from IL10R-deficient mice closely matched gene expression profiles obtained from MΦs isolated from control mice, and lacked strong pro-inflammatory gene expression. These results suggest that LP MΦs of yolk sac origin maintain an anti-inflammatory state even in the absence of IL10R signaling. While the absence of an inflammatory phenotype in IL10R-deficient LP MΦs prior to weaning may be explained by exogenous factors such as difference in the microbiota or diet prior to and after weaning, it is also possible that this phenotype is a result of fundamental differences in MΦ lineages. Understanding the basis for differences in the phenotype of IL10R-deficient LP MΦs prior to and after weaning is an important line of future investigation, and should such studies suggest that there are intrinsic differences in the phenotype of IL10R-deficient LP MΦs prior to weaning, detailed analysis of transcriptional regulation in infant IL10R-deficient LP MΦs could reveal IL10R-independent pathways that inhibit MΦ inflammatory responses. Identification of these pathways could lead to novel therapeutic approaches to patients with defects in IL10R signaling.

While our study clearly demonstrates the importance of IL10R signaling on MΦs to prevent intestinal inflammation in the infant mouse, the exact mechanisms through which IL10 signaling inhibits inflammatory gene expression in LP MΦs is not completely understood. We have recently shown that IL10 rapidly inhibits LPS-induced enhancer activation in bone marrow-derived MΦs (*Conaway et al., 2017*), but do not yet know whether the absence of IL10R signaling influences the function of enhancers in LP MΦs. Further, it remains unclear whether the critical function of IL10R is to directly inhibit inflammatory gene expression in MΦs or alternatively to promote production of a factor with a broad inhibitory function. Although it is clear that IL10R signaling strongly inhibits a group of inflammation-induced genes, the ability of a relatively small number of adoptively transferred IL10R-sufficient MΦs to suppress the development of colitis in *Rag2*$^{-/-}$*Il10rb*$^{-/-}$ mice reconstituted with CD4$^{+}$ T cells (*Shouval et al., 2014b*) suggests that WT MΦs exert a dominant effect over IL10R-deficient MΦs. This type of effect is more compatible with a requirement for IL10R to induce production of an inhibitory factor than an intrinsic requirement to prevent inflammatory gene expression. Further studies using mixed radiation chimeras with congenically marked control and IL10R-deficient MΦs will be necessary to fully understand this issue.

Collectively, our data indicate that IL10R signaling in MΦs is pivotal in confining a microbiota-driven inflammatory response beginning at the third week of life. Alterations in colonic LP anti-inflammatory MΦ phenotype is a key early event initiating this inflammatory cascade. While further directed experiments are still needed to address the primary function of IL10R in resident colonic MΦs during infant development, it is enticing to speculate that the intestinal inflammation observed in infants lacking IL10R signaling may be based on the inability of host MΦs to control the inflammatory potential of the expanding intestinal microbiota. In short, our data has uncovered a critical 'window of opportunity' to further dissect the mechanism(s) leading to the development of infantile colitis.

## Materials and methods

### Mice

*Il10rb*[-/-] (*Il10rb*[tm1Agt]) (RRID: MGI:3603437) (*Shouval et al., 2014b*) mice were maintained on the 129SvEv background. *Il10ra*[fl/fl] (C.129P2-*Il10r*[tm1(flox)Greifswald]) (*Pils et al., 2010*), *Il10ra*[-/-] (*Pils et al., 2010*), *Lyz2*[Cre] (B6.129P2-*Lyz2*[tm1(cre)Ifo]/J), also known as LysM[Cre] (RRID: IMSR_JAX:004781) (The Jackson Laboratory, Bar Harbor, ME), and *Cdcs1*[+/+] (B6.C3Bir-*Cdcs1* (BC-R)) (*Bleich et al., 2010*) mice were maintained on the C57BL/6 background. All mice were housed in a specific pathogen-free animal facility at Boston Children's Hospital. Controls consisted of co-housed littermates of the appropriate genotype. Both male and female mice were used throughout this study. The 2 week age of mice represented day 14–15. The 3 week represented mostly day 21, whilst mice aged between day 21–28 were sometimes grouped as 3–4 week old. All experiments were conducted following approval from the Animal Resources at Children's Hospital, per regulations of the Institutional Animal Care and Use Committees (IACUC, Assurance number: A3303-01).

### Reagents and antibodies

The antibiotics vancomycin, polymyxin B, ampicillin, and neomycin were obtained from Gold Biotechnology, Inc. (St. Louis, MO), and metronidazole was from Sigma-Aldrich Corp. (St. Louis, MO). Liposomal clodronate and control liposomes were obtained from Encapsula NanoSciences LLC (Brentwood, TN). The following reagents and anti-mouse antibodies were used for flow cytometry: Zombie Violet[TM] fixable viability kit (BioLegend, San Diego, CA), CD45 (clone 30-F11, BioLegend), CD11b (clone M1/70, BioLegend), CD11c (clone N418, BioLegend), CD103 (clone 2E7, eBioscience/ Thermo Fisher Scientific, Waltham, MA), CD64 (clone X54-5/7.1, BioLegend), I-A/I-E (MHCII, clone M5/114.15.2, BioLegend), Ly6C (clone HK1.4, BioLegend), CD3e (clone 145–2 C11, BioLegend), CD4 (clone GK1.5, BioLegend), FoxP3 (clone FJK-16s, eBioscience), IFNγ (clone XMG1.2, BioLegend), IL17A (clone TC11-18H10.1, BioLegend), CD127 (clone A7R34, BioLegend), CD90.2 (Thy1.2, clone 30-H12, BioLegend), and RORγt (clone B2D, eBioscience).

### Spontaneous colitis monitoring and scoring

High-resolution endoscopy of colons (Karl Storz, Tuttlingen, Germany) was performed under anesthesia and findings were graded as previously described (*Shouval et al., 2014b*; *Becker et al., 2006*). Transverse sections prepared from distal colons were H and E stained and scored in a blinded fashion by the board-certified veterinary pathologist (V. B.) using the scoring criteria adapted from Rogers and Houghton (*Rogers and Houghton, 2009*). Briefly, scores ranged from 0 to 24, based on inflammation (0–4), edema (0–4), epithelial defects (0–4), crypt atrophy (0–4), hyperplasia (0–4), and dysplasia (0–4).

### Isolation of LP cells

LP immune cells were prepared as we reported recently (*Shouval et al., 2014b*, *2016*). Briefly, colons were stripped of epithelial cells by performing agitation in 10 mM EDTA for 20 min at 37°C twice before digestion in collagenase VIII (Sigma-Aldrich) for 30–45 min at 37°C. Undigested tissue were disrupted by repeated flushing through a 10 ml syringe and then subjected to additional 10 min incubation in shaking conditions. Single cell suspensions were filtered and stained for flow cytometry.

### Quantitative real-time PCR

Total RNA was extracted from whole colons using TRIzol reagent (Invitrogen, Grand Island, NY) per the manufacturer's instructions. cDNA was reverse transcribed from 1 μg total RNA using iScript Select cDNA Synthesis Kit (Bio-Rad Laboratories, Hercules, CA). Analyses of transcripts were performed using iQ SYBR Green or SsoAdvanced Universal SYBR Green Supermixes on a CFX96 Real-Time System (Bio-Rad). Cytokine/chemokine transcripts were normalized against hypoxanthine-guanine phosphoribosyl transferase (HPRT) or glyceraldehyde 3-phophate dehydrogenase (GAPDH), and relative expression was quantified with the $2^{-\Delta Ct}$ method (*Powell et al., 2012*).

## Flow cytometry

LP cells were acquired using FACS Canto II flow cytometer (BD Biosciences, San Jose, CA) and analyzed using FlowJo v9 and v10 (Treestar, Ashlan, OR). A 1:1 mix of mouse and rat sera was used at 20% concentration as Fc blocking reagent. For intracellular cytokine staining, cells were stimulated for 4.5 hr with 500 ng/mL ionomycin and 50 ng/mL PMA in the presence of 1 μg/mL of GolgiStop containing Monensin (BD Biosciences). Cell were then fixed, permeabilized (BD Biosciences), and stained for intracellular proteins. For FoxP3 staining, we used the fixation/permeabilization kit by eBioscience.

## Antibiotics administration

The broad-spectrum antibiotic cocktail contained metronidazole (1 mg/ml), vancomycin (0.5 mg/ml), ampicillin (1 mg/ml), and neomycin (1 mg/ml). Polymyxin B was used at 100 μg/ml and vancomycin at 0.5 mg/ml. Antibiotics were provided to pregnant $Il10rb^{-/-}$ or $Il10rb^{+/-}$ dams ad libitum in the drinking water starting on day E16-18. Pups obtained from these antibiotic-treated cages were weaned at 3 weeks into cages with the respective antibiotics before euthanasia at 4 weeks.

## Clodronate-mediated macrophage depletion

Liposomal preparation of clodronate or control liposomes were injected intraperitoneally to 3 weeks old mice at 50 mg/kg body weight three times within a week. Mice were then euthanized and evaluated for LP MΦ depletion and colitis development.

## RNA sequencing and computational analysis

10,000–40,000 LP MΦs (P3/P4 fraction) from 1 week (day 10–11), 3 week (day 21–26), and 12-week-old $Il10rb^{-/-}$ and $Il10rb^{+/-}$ mice were FACS-sorted (gates: CD45$^+$CD11b$^+$CD11c$^{int}$ CD103$^-$CD64$^+$-Ly6C$^-$MHCII$^+$ (*Figure 2—figure supplement 1*)) directly in RLT lysis buffer (Qiagen, Waltham, MA) and mRNA was extracted using the Qiagen RNeasy Micro kit. Oligo dT beads were used to enrich for mature polyA(+) transcripts. RNA-Seq was carried out at the Molecular Biology Core Facility (MBCF) of Dana-Farber Cancer Institute, Boston using the Nextseq Standard Program SE75, and reads were aligned to mm10 using TopHat with default settings, and DESeq was used to generate normalized counts for each gene. Only the genes that had normalized counts $\geq$ 50 in at least one of the total six groups were further analyzed. Heatmaps were generated using 'R' studio from genes that were differentially-expressed (>3 fold) between LP MΦ of 3 week old $Il10rb^{-/-}$ and $Il10rb^{+/-}$ mice. All data has been deposited in NCBI's Gene Expression Omnibus and are accessible through GEO Series accession number GSE97875.

## Fecal DNA preparation and pyrosequencing

Fecal pellets were collected from groups of 10–12 $Il10rb^{-/-}$ and littermate $Il10rb^{+/-}$ mice starting at day 17 and then every other day until day 28, followed by every week until 14 weeks. Samples were snap frozen and DNA was subsequently extracted by using PowerSoil DNA Isolation Kit (MoBio/Qiagen, Carlsbad, CA) according to the manufacturer's instructions. Pyrosequencing was performed as described in detail (*Wang et al., 2015*) with some modifications. Briefly, the V4 region of the 16S rRNA gene was amplified by PCR using bar-coded universal primers 354F and 926R (Invitrogen/ThermoFisher Scientific, Waltham, MA). DNA emulsion PCR and 454 DNA sequencing were performed at the BCM Human Genome Sequencing Center in Houston, TX. Sequencing was performed using the 454/Roche B sequencing primer kit in the Roche Genome Sequencer GS-FLX Titanium platform.

## 16S rRNA data processing and analysis

Raw sequence reads were processed using the FASTX-Toolkit (http://hannonlab.cshl.edu/fastx_toolkit/) and Qiime 1.9.1 (*Caporaso et al., 2010*) with default settings. Quality filtering was done with split_libraries_fastq.py, and pick_open_reference_otus.py was used to pick operational taxonomic units (OTUs) against the Greengenes database (*DeSantis et al., 2006*; *McDonald et al., 2012*). The 16S data was rarefied to 40,000 sequences per sample. Alpha- and beta-diversity metrics were assessed using core_diversity_analyses.py. To characterize the *Helicobacter* OTU, phylogenetic placement using the pplacer package (10.1186/1471-2105-11-538) was performed against a

reference tree constructed using full-length or near full-length 16S rDNA sequences from eight known murine *Helicobacter* species.

## Statistical analysis

Statistical analyses were performed with GraphPad Prism 6.0 software unless stated otherwise. Differences between groups were compared using the 2-way ANOVA followed by Bonferroni or Tukey's multiple comparison tests, or Mann-Whitney *U* test, as indicated. Statistical significance was defined as *p* value was less than 0.05. *, **, ***, **** refer to p<0.05; p<0.01; p<0.001; and p<0.0001, respectively.

## Acknowledgements

Authors are grateful to Prof. Matthias Mack, Universitätsklinikum Regensburg, Germany, for the kind gift of anti-CCR2 (MC21) antibody. Authors would like to thank the members of the Snapper and the Horwitz laboratories for helpful discussions and critical reading of the manuscript.

## Additional information

### Funding

| Funder | Grant reference number | Author |
| --- | --- | --- |
| Crohn's and Colitis Foundation of America | RFA381023 | Naresh S Redhu |
| Canadian Institutes of Health Research | 201411MFE-339308-254788 | Naresh S Redhu |
| National Institutes of Health | T32-OD010978-26 | James G Fox |
| Leona M. and Harry B. Helmsley Charitable Trust | | Scott B Snapper |
| Wolpow Family Chair in IBD Research and Treatment | | Scott B Snapper |
| National Institutes of Health | R01-OD011141 | James G Fox |
| National Institutes of Health | P30-ES002109 | James G Fox |
| National Institutes of Health | R01-AI00114 | Bruce H Horwitz |

The funders had no role in study design, data collection and interpretation, or the decision to submit the work for publication.

### Author contributions

NSR, Conceptualization, Data curation, Formal analysis, Investigation, Writing—original draft, Writing—review and editing; VB, AT, Data curation, Formal analysis, Investigation, Writing—review and editing; EAC, CW, MF, Investigation, Writing—review and editing; DSS, Writing—review and editing; JAG, ABi, NL, Formal analysis, Investigation, Writing—review and editing; WM, ABl, Resources, Writing—review and editing; GKG, JGF, Supervision, Methodology, Writing—review and editing; LB, Supervision, Writing—review and editing; SBS, Conceptualization, Resources, Formal analysis, Funding acquisition, Investigation, Methodology, Writing—review and editing; BHH, Conceptualization, Formal analysis, Supervision, Funding acquisition, Investigation, Methodology, Writing—original draft, Project administration, Writing—review and editing

### Author ORCIDs

Naresh S Redhu, http://orcid.org/0000-0002-0264-9925
Amlan Biswas, http://orcid.org/0000-0003-4299-1001
Georg K Gerber, http://orcid.org/0000-0002-9149-5509
Bruce H Horwitz, http://orcid.org/0000-0002-8123-8728

## Ethics

Animal experimentation: All experiments were conducted following approval from the Animal Resources at Children's Hospital, per regulations of the Institutional Animal Care and Use Committees (IACUC assurance number A3303-01).

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
