## [Decision Letter]

Thank you for submitting your article "Macrophage dysfunction initiates colitis during weaning of infant mice lacking the interleukin-10 receptor" for consideration by *eLife*. Your article has been favorably evaluated by Tadatsugu Taniguchi (Senior Editor) and three reviewers, one of whom is a member of our Board of Reviewing Editors. The following individual involved in review of your submission has agreed to reveal their identity: Christoph Mueller (Reviewer #2).

The reviewers have discussed the reviews with one another and the Reviewing Editor has drafted this decision to help you prepare a revised submission.

Summary:

The paper is a valuable detailed analysis of the early changes of the IL-10 receptor (IL-10R) deficiency in mice and its pro-inflammatory consequences around the time of weaning germane to the monogenic IL-10R causes of chronic intestinal inflammation in young children. These include colonic pro-inflammatory cytokine gene expression and compensatory IL-10 expression. There are increased P1 and P2 macrophage populations in IL-10Rbeta-deficient animals, whereas their P3/4 populations have an inflammatory signature. The inflammatory phenotype is shown to be macrophage-dependent through depletion using clodronate liposomes or macrophage-specific deletion of the IL-10Ralpha receptor. Broad-spectrum antibiotics to reduce the intestinal microbial biomass ameliorate the inflammatory phenotype, with polymyxin B alone (targeting Gram-negatives) being more effective than vancomycin (targeting Gram-positives). No effects of the IL-10R host trait were seen on microbiota α or β diversity, or Helicobacter levels, although the consortial composition required to drive inflammation remains unclear.

Essential revisions:

1) A major point was the detailed microbial requirements to drive the inflammatory phenotype. Is exposure to the molecular signature from Gram-negative microbes (i.e. is LPS generically sufficient from any Gram-negative) or is it triggered by colonization by a context-dependent pathobiont beyond the resolution of the ordination analysis?

It was also noted that in a previous article (reference 20) on a related topic, the kinetics of intestinal inflammation observed in *IL10rb^-/-^* mice was more attenuated (reference 20, page 709: "we evaluated *Il10rb^-/-^* mice at 5 weeks of age that lacked any clinical (data not shown), endoscopic, or histologic signs of intestinal inflammation (Figure 3)" and in that study (reference 20) for rapid colitis induction adoptive transfer of colitogenic CD4 T cells into *IL10rb^-/-^ Rag2^-/-^* mice was used. In the present study, the *IL10rb-* and *IL10rb^+/-^* mice were found to be colonized with *Helicobacter sp.* around day 17 of age as assessed by 16S RNA pyrosequencing. Whilst formal rederivation to monocolonised or *Helicobacter*-free status is not necessarily considered within the current scope, please address whether differences in the microbial composition (notably, for *Helicobacter sp.*) might account for these differences between the two reports. The caveats to address how microbiota constraints may be responsible for the phenotype need to be included in the Discussion?

2) Comment on whether you have examined the relevance of the mechanisms of inhibition of inflammatory gene transcription in BMDMs by IL-10 (previous publication JI 2017, 198, 2906-2915) in the context of macrophages in this model.

3) Specifically clarify the question of chromosome trait dominance determination. The text and figure legends are unclear about exactly when littermates or wild-type age-matched mice were used (compare the subsection “Mice” of Materials and methods and Figure legends 1 and 2).

4) *IL10ra^-/-^* and *IL10rb^-/-^* mice on a different genetic background were used. *IL10Rb* deficient mice were maintained on a 129SvEv background, while *IL10ra* deficient mice were used on a *Cdcs1*- congenic, C57BL/6 background. While the colitogenic locus identified in 129SvEv mice (Hiccs) phenocopies most of the effects seen with the *Cdcs1* locus, Hiccs overlaps only with the most telomeric sublocus of *Cdcs1* (Boulard et al., J Exp Med. 209: 1309-24; 2012). Hence, it remains to be seen whether differences seen in *IL10ra*, and *IL10rb* deficient mice are due to the differential binding of other ligands to *IL10ra*, and *IL10rb*; a distinct composition of the intestinal microbiota; notably, of Helicobacter species, and/or the genetic background of the mouse strains. These possibilities need to be considered in the interpretation of the data.

5) While the authors comment that other work has suggested that CCR2 may not be required for myeloid cell infiltration in some models of colitis, neonatal mice have not been examined in this way. In view of the marked infiltration by monocytes reported here, it should be clear if the authors have conducted any experiments on a potential role for CCR2 in their models.

6) The authors do not have direct support for their conclusion that embryonically and monocyte-derived macrophages may have intrinsic differences in their regulation by IL10. This idea is based only on comparing macrophages from mice before (embryonic) and after (monocyte derived) the onset of the changes in the microbiota at weaning. Thus, the differences seem likely to reflect the combined effect of microbial stimulation and the absence of IL10 control, rather than an effect of the source of macrophages.

---

## [Author Response]

Essential revisions:

1) A major point was the detailed microbial requirements to drive the inflammatory phenotype. Is exposure to the molecular signature from Gram-negative microbes (i.e. is LPS generically sufficient from any Gram-negative) or is it triggered by colonization by a context-dependent pathobiont beyond the resolution of the ordination analysis?

The reviewer is raising an important issue regarding the nature of the microflora that drive the onset of infantile colitis in our model system. While our experiments with antibiotics demonstrate that those with specificity towards Gram-negative organisms can inhibit the development of inflammation, we have not tested individual organism or defined communities in this model, or whether the introduction of a gram-negative derived molecular pattern such as LPS would induce colitis in the IL10R-deficient infant mice described here. However, it is well described that members of the gram-negative enterohepatic *Helicobacter* species (EHS) including *H. hepaticus* function as pathobionts and are required for the development of colitis in many immunodeficient strains, including IL10-deficient mice (Kullberg MC et al., Infect Immun 1998; Fox JG et al., Mucosal Immunol 2011). While IL10-deficient mice mono-colonized with *H. hepaticus* do not develop colitis, colonization with both *H. hepaticus* and *Lactobacillus reuteri* led to severe typhlocolitis in germ-free IL10-deficient mice (Whary MT et al., Immunology 2011). In addition, we have recently presented data demonstrating that germ-free Was-deficient mice develop colitis when colonized by both *H. bilis* (an EHS related to *H. hepaticus*) and altered Schaedler’s flora (ASF), but not when colonized by either component alone (Tsou et al., Gastroenterology supplement, April 2017, S111). Collectively, these studies suggest that colonization with enterohepatic *Helicobacter* in association with other minimal microbial communities (ASF or *L. reuteri*) is sufficient to drive intestinal inflammation in susceptible murine strains. Given that we have observed marked increases in colonization with *H. ganmani* (another Gram-negative EHS related to *H. hepaticus*) in both WT and IL10R-deficient mice between 2 and 3 weeks of age, we hypothesize that colonization by a context-dependent Gram-negative pathobiont such as *H. ganmani*, in addition to other minimal microbial communities is necessary for the development of inflammation in the infant model presented here. Future experiments will be necessary to unambiguously identify the specific microflora necessary to drive infant colitis in this model. We have revised the Discussion in consideration of these issues (fourth paragraph).

It was also noted that in a previous article (reference 20) on a related topic, the kinetics of intestinal inflammation observed in IL10rb^-/-^ mice was more attenuated (reference 20, page 709: "we evaluated Il10rb^-/-^ mice at 5 weeks of age that lacked any clinical (data not shown), endoscopic, or histologic signs of intestinal inflammation (Figure 3)" and in that study (reference 20) for rapid colitis induction adoptive transfer of colitogenic CD4 T cells into IL10rb^-/-^ Rag2^-/-^ mice was used. In the present study, the IL10rb- and IL10rb^+/-^ mice were found to be colonized with Helicobacter sp. around day 17 of age as assessed by 16S RNA pyrosequencing. Whilst formal rederivation to monocolonised or Helicobacter-free status is not necessarily considered within the current scope, please address whether differences in the microbial composition (notably, for Helicobacter sp.) might account for these differences between the two reports. The caveats to address how microbiota constraints may be responsible for the phenotype need to be included in the Discussion?

As pointed out by this reviewer, we recognize that differences in microbial composition may account for differences in time of disease onset when comparing this study and our prior study. We note that the animal housing conditions have changed over time. In contrast to our previous report where *Il10rb^-/-^* and WT mice were bred separately and were not co-housed, the mice in our current study were generated with an independent breeding strategy where all mice were co-housed littermate controls. In addition, the mice in our current study were confirmed positive for *Helicobacter sp.* whereas we are unable to verify the *Helicobacter* status of mice from our previous study, as the stool samples from those mice are unavailable. Finally, approximately 15-20% of *Il10rb^-/-^* mice in our current study demonstrate delayed onset of histologic colitis, whereas the altered LP macrophage phenotype is observed in 100% of the mice beginning at 3 weeks. In the previous study, mice that lacked colitis at 5 weeks were specifically selected to demonstrate that LP macrophage dysfunction precedes the development of microscopic colitis, however, we did not intend to suggest that all mice lacked colitis at 5 weeks of age. We have revised our Discussion based on these issues, and intend to pursue this further in follow-up studies (fourth paragraph).

2) Comment on whether you have examined the relevance of the mechanisms of inhibition of inflammatory gene transcription in BMDMs by IL-10 (previous publication JI 2017, 198, 2906-2915) in the context of macrophages in this model.

We have recently demonstrated that IL10 rapidly inhibits LPS-induced enhancer activation in bone marrow-derived macrophages (Conaway E et al.,J Immunol 2017). While we have not yet directly examined mechanisms responsible for altered gene expression in LP macrophages isolated from IL10R-deficient mice, we believe that this is a priority for future mechanistic-based research in this area. In response to this comment we have broadened the Discussion of this issue (seventh paragraph).

3) Specifically clarify the question of chromosome trait dominance determination. The text and figure legends are unclear about exactly when littermates or wild-type age-matched mice were used (compare the subsection “Mice” of Materials and methods and Figure legends 1 and 2).

All aged-matched mice used in these studies were littermates with one exception. Five wild-type age-matched non-littermate mice were combined with 5 *Il10rb^+/-^* mice to make up the control group for the 3 week time point shown in the manuscript (original submission, Figure 1). In our revised version, we have removed the 5 3-week old WT control mice in this figure. While this results in some minor differences in statistical significance at 3 wks (*p* values increased from *0.01* to *0.09* for *Il12b, 0.05 to 0.17 for Il1b*, and *0.08* to *0.38* for *Tnf*), these small changes do not influence our overall conclusions. The Materials and methods section has been updated to reflect this (subsection “Mice”).

4) IL10ra^-/-^ and IL10rb^-/-^ mice on a different genetic background were used. IL10Rb deficient mice were maintained on a 129SvEv background, while IL10ra deficient mice were used on a Cdcs1- congenic, C57BL/6 background. While the colitogenic locus identified in 129SvEv mice (Hiccs) phenocopies most of the effects seen with the Cdcs1 locus, Hiccs overlaps only with the most telomeric sublocus of Cdcs1 (Boulard et al., J Exp Med. 209: 1309-24; 2012). Hence, it remains to be seen whether differences seen in IL10ra, and IL10rb deficient mice are due to the differential binding of other ligands to IL10ra, and IL10rb; a distinct composition of the intestinal microbiota; notably, of Helicobacter species, and/or the genetic background of the mouse strains. These possibilities need to be considered in the interpretation of the data.

While we believe that our data strongly support our contention that the IL10R itself has a function that is necessary to prevent the onset of colitis between the second and third week of life in infant mice, we have not directly compared the phenotype of *Il10rb^-/-^* and *Il10ra^-/-^* mice. Nonetheless, we agree that there may be differences in the phenotype of the *Il10rb^-/-^* and *Il10ra^-/-^* mice and such difference could be based on differences in the function of these two receptor components, genetic background, and/or intestinal microbial composition. We have now included these issues in the interpretation of experimental results (subsection “IL10Rα chain is essential to prevent inflammation in the developing infant colon”).

5) While the authors comment that other work has suggested that CCR2 may not be required for myeloid cell infiltration in some models of colitis, neonatal mice have not been examined in this way. In view of the marked infiltration by monocytes reported here, it should be clear if the authors have conducted any experiments on a potential role for CCR2 in their models.

We appreciate this important comment. Indeed, we have considered the potential role of CCR2 in driving colonic inflammation in neonatal *Il10rb^-/-^* mice. We found that treatment of 3 week old *Il10rb^-/-^* mice with the anti-CCR2 antibody (MC21, gift of Dr. Matthias Mack, Germany) markedly decreased infiltration of Ly6C^+^ cells (P1 and P2; Figure 9), but did not decrease the number of P3/P4 macrophages. However, we were unable to detect a significant difference in the colonic expression of inflammatory genes including *Il12b, IFNγ, Tnf*, and *Il1b* between isotype and MC21 antibody-treated *Il10rb^-/-^* mice (p>0.05, Figure 9). Given that the P3/P4 macrophages in infant *Il10rb^-/-^* mice exhibit a proinflammatory phenotype (Figure 3), this is consistent with the hypothesis that P3/P4 macrophages are the key driver of colonic inflammation in this model. Nonetheless, the CCR2 antibody experiment has an important caveat that the depletion was transient and we do not know if this was sufficient to interfere with the inflammatory process in IL10R deficiency setting. Therefore, given that our preliminary data on CCR2 depletion is not definitive, it would be ideal to breed our *Cdcs1^+/+^Il10ra^-/-^* mice with *Ccr2^-/-^* mice, which although beyond the scope of our current study, will remain an important question for our future investigations. We have noted the results of this preliminary experiment in the Discussion section of this revised manuscript (fifth paragraph).

Author response image 1.CCR2 blocking does not inhibit inflammatory gene expression in the infant *Il10rb^-/-^* mice.(**A**) Flow cytometry analyses of colonic lamina propria macrophages of indicated mice after 5 days of treatment (20μg/mouse/day, i.p.) with anti-CCR2 (MC21) or rat IgG2b, κ (isotype). (**B**) Graphical summary of the frequency and numbers of colonic LP macrophage subsets gated on CD45^+^CD11b^+^CD11c^int^CD103^-^CD64^+^ cells. (**C**) Graphical summary of gene expression by qRT-PCR from the colons of mice indicated above. *p<0.05, **p<0.01, ***p<0.001, Mann-Whitney *U* test. Data shows Median + IQR, each data point indicates individual mouse.**DOI:**
http://dx.doi.org/10.7554/eLife.27652.019

6) The authors do not have direct support for their conclusion that embryonically and monocyte-derived macrophages may have intrinsic differences in their regulation by IL10. This idea is based only on comparing macrophages from mice before (embryonic) and after (monocyte derived) the onset of the changes in the microbiota at weaning. Thus, the differences seem likely to reflect the combined effect of microbial stimulation and the absence of IL10 control, rather than an effect of the source of macrophages.

We agree with the reviewers that it is difficult to know whether the lack of an inflammatory phenotype in IL10R-deficient LP MΦs prior to weaning is truly due to differences in site of origin, or conversely due to the absence of specific inflammatory microflora. We have raised these considerations in the revised Discussion (sixth paragraph).